# SWARM Parallelism: Training Large Models Can Be Surprisingly Communication-Efficient

## Abstract

Many deep learning applications benefit from using large models with billions of parameters. These models can only be trained with specialized distributed training algorithms that require low-latency and high-bandwidth interconnect. As a result, large models are typically trained in dedicated GPU clusters that can be extremely costly to deploy and operate. In contrast, there are more affordable distributed training setups, such as using cheap "preemptible" instances or pooling together existing resources from multiple regions. However, both these setups come with unique challenges that make it impractical to train large models using conventional model parallelism. In this work, we carefully analyze these challenges and find configurations where *training larger models becomes less communication-intensive*. Based on these observations, we propose SWARM Parallelism[1] — a model-parallel training algorithm designed for swarms of poorly connected, heterogeneous unreliable devices. SWARM creates temporary randomized pipelines between available nodes that are rebalanced in case of failure. To further reduce the network usage of our approach, we develop several compression-aware architecture modifications and evaluate their tradeoffs. Finally, we combine our insights to train a large Transformer language model with 1.1B shared parameters ($\approx$13B before sharing) on a swarm of preemptible T4 GPUs with less than 400Mb/s network throughput.

## 1 Introduction

For the past several years, the deep learning community has been growing ever more reliant on large pretrained models. Perhaps the easiest example of this trend is natural language processing, where the parameter count of models grew from hundreds of millions (Vaswani et al., 2017; Radford et al., 2018; Devlin et al., 2019) to billions (Narayanan et al., 2021; Rosset; Raffel et al., 2020; Wang & Komatsuzaki, 2021; Sun et al., 2021) to hundreds of billions (Brown et al., 2020; Lepikhin et al., 2020; Fedus et al., 2021) with consistent gains in quality (Kaplan et al., 2020). Likewise, many models in computer vision are reaching the billion-parameter scale (Henighan et al., 2020; Ramesh et al., 2021; Zhai et al., 2021; Riquelme et al., 2021; Dai et al., 2021; Dhariwal & Nichol, 2021).

At this scale, the model no longer fits into a single accelerator and requires specialized training algorithms that partition model parameters across devices (Krizhevsky et al., 2012; Dean et al., 2012). While these model-parallel algorithms use different partitioning strategies, they all share the need to perform intensive device-to-device communication (Narayanan et al., 2019; 2021). Furthermore, if a single device fails, it will cause the entire training to break down. As a result, model-parallel algorithms are typically deployed in dedicated HPC[2] clusters or supercomputers (Shoeybi et al., 2019; Rajbhandari et al., 2020; Narayanan et al., 2021).

This kind of infrastructure is notoriously expensive to build and operate, available only to few well-funded universities and large corporations (Larrea et al., 2019; Strohmaier et al., 2021; Langston, 2020). Most researchers, especially in developing nations, simply cannot afford to run the necessary experiments to properly evaluate their ideas. This ultimately limits the scientific progress for many important research areas, such as solving NLP problems in "non-mainstream" languages.

---

[1]Stochastically Wired Adaptively Rebalanced Model Parallelism
[2]HPC stands for high performance computing, typically in supercomputers.

Several recent works propose more cost-efficient distributed training strategies that use fleets of temporary "preemptible" instances that can be dynamically allocated in regions with low demand for hardware and electricity, making them 2–10 times cheaper than their dedicated counterparts (Harlap et al., 2017). Another solution is to train in "collaborations" by pooling together preexisting resources or using the help of volunteers (Diskin et al., 2021; Atre et al., 2021; Ryabinin & Gusev, 2020). However, training in either of those setups requires specialized algorithms that can adapt to the changing number of workers, utilize heterogeneous devices and recover from hardware and network failures. While there are several practical algorithms for unreliable hardware (Kijsipongse et al., 2018; Lin et al., 2020; Ryabinin et al., 2021), they can only train relatively small models that *fit into the memory of the smallest device*. This limits the practical impact of cost-efficient strategies, as most computationally demanding workloads typically train models with billions of parameters.

In this work, we aim to find a practical way of training large models using unreliable heterogeneous devices and slow interconnect. Our contributions can be summarized as such:

- We carefully analyze the existing model-parallel training techniques and formulate the "Square-Cube Law" of distributed training: a counterintuitive observation that, for some methods, *training larger models can actually decrease the network overhead*.

- We develop **SWARM parallelism**: a decentralized model-parallel algorithm[3] that replaces traditional pipelines with randomized temporary routing between swarms of peers, allowing it to operate with heterogeneous devices and recover from node failures. To the best of our knowledge, this is the first general-purpose algorithm capable of billion-scale training on heterogeneous unreliable devices with slow interconnect.

- We study compression-aware architecture variants that utilize bottlenecks and activation sparsity to significantly reduce the network bandwidth requirements. We evaluate the performance and quality tradeoffs of two different methods and find that our best compression technique, that uses the maxout activation function for compression, reduces bandwidth requirements by 4x while degrading performance by only 4–8%.

- Combining SWARM parallelism and compression-aware techniques, we demonstrate that it is possible to train a billion-scale Transformer language model with high training throughput on preemptible low-power T4 GPUs and $< 400$Mb/s network.

## 2 BACKGROUND & RELATED WORK

### 2.1 MODEL-PARALLEL TRAINING

Deep learning community came up with multiple algorithms for training large models. Most of them work by somehow dividing the model between multiple workers, which is known as model parallelism. The exact way in which these algorithms divide the model significantly affects their training performance and the maximum model size they can support.

**Traditional model parallelism.** Historically, the first general strategy for training large models was to assign each device to compute a subset of each model layer (e.g. a subset of neurons), then communicate activations between each other (Krizhevsky et al., 2012); see surveys by Ben-Nun & Hoefler (2019) and Tang et al. (2020) for a more comprehensive overview. Since each device stores a fraction of layer parameters, this technique can train extremely wide layers that would not fit into a single GPU. However, applying traditional model parallelism to deep models comes with a significant performance penalty, as it requires all-to-all communication after each layer. As a result, while intra-layer parallelism schemes are still widely used (Shazeer et al., 2018; Rajbhandari et al., 2020), they are usually applied within one physical server in combination with other strategies (Krizhevsky, 2014; Chilimbi et al., 2014; Jia et al., 2019; Narayanan et al., 2021).

**Pipeline parallelism** circumvents the need for expensive all-to-all communication by assigning each device with one or several full layers (Huang et al., 2019). During the forward pass, each stage applies its subset of layers to the inputs supplied by the previous stage, then sends the outputs of the final layer to the next stage. For backpropagation, this process is reversed, with each pipeline stage passing the gradients to the same device that previously supplied it with input activations.

---

[3]The code for our experiments can be found at `github.com/iclr2022-submit/swarm`

To better utilize the available devices, the pipeline must process multiple micro-batches per step, allowing each stage to run in parallel on a different batch of inputs. In practice, the number of micro-batches is limited by the available device memory, which results in the reduced device utilization when processing first and last micro-batches, which is known as the "bubble" overhead (Huang et al., 2019). To combat this issue, subsequent studies propose using activation checkpointing, interleaved scheduling, and even asynchronous training (Narayanan et al., 2019; 2021; Huang et al., 2019; Shoeybi et al., 2019; Yang et al., 2019). Another limitation of pipeline parallelism is that it needs to evenly split the compute load across the pipeline stages (Narayanan et al., 2019), which makes it difficult to apply when model layers have uneven computational cost.

Finally, there are two types of algorithms that can be used to train large models: data parallelism with dynamic parameter loading (Rajbhandari et al., 2020; Pudipeddi et al., 2020; Ren et al., 2021) and layer-dependent algorithms such as distributed Mixture-of-Experts (Jacobs et al., 1991; Shazeer et al., 2017; Lepikhin et al., 2020; Fedus et al., 2021). We review these algorithms in Appendix A and compare our approach with pipeline parallelism and offloading in Appendices E and F.

## 2.2 DISTRIBUTED TRAINING OUTSIDE HPC

The distributed training techniques described in Section 2.1 are designed for clusters of identical devices with rapid and reliable communication, making them a natural fit for the High-Performance Computing (HPC) environment. As we discussed earlier, this infrastructure is not always available or affordable. A more cost-efficient alternative is to use "preemptible" instances (Li et al., 2019; Zhang et al., 2020; Harlap et al., 2017) or volunteer computing (Diskin et al., 2021; Kijsipongse et al., 2018; Atre et al., 2021; Ryabinin & Gusev, 2020). However, these environments are more difficult for distributed training, as each machine can leave training abruptly due to hardware failure or spikes in demand. Furthermore, since there is a limited number of available instances per region, training at scale often requires operating in multiple geographic regions or using multiple instance types.

To summarize, there are five main limitations for distributed training in non-HPC environments: (1) nodes joining and leaving over time, (2) node and network failures, (3) uneven performance from heterogeneous devices, and, finally, (4) low network bandwidth and (5) high latency between regions.

In order to train with a dynamic number of workers, deep learning practitioners came up with elastic training algorithms (TorchElastic; ElasticHorovod). If one peer leaves or fails midway through training, these algorithms dynamically re-balance the load between the remaining devices and continue the training (Harlap et al., 2017; Ryabinin et al., 2021). If new devices join midway through training, they download the latest training parameters from their peers and train alongside them.

Another line of research tackles distributed training on heterogeneous devices with uneven performance. One way to solve this problem is to use asynchronous training, where devices compute gradients at their own pace and aggregate them using either a parameter server (Recht et al., 2011; Kijsipongse et al., 2018) or a decentralized communication network (Lian et al., 2017). This idea allows asynchronous algorithms to fully utilize each individual device, but may reduce the convergence rate due to using "stale" gradients (Recht et al., 2011; Aji & Heafield, 2019). Finally, several studies (Li et al., 2020; Ryabinin et al., 2021; Ren et al., 2021; Diskin et al., 2021) propose hybrid techniques that remove some synchronization bottlenecks while maintaining per-iteration convergence.

However, all of these use cases rely on data-parallel training and thus share a fundamental limitation: each computing node must be able to run the entire model. In contrast, the most compute-intensive tasks require training models with billions of parameters, which is way outside both GPU and host memory of most low-end GPUs. Unlike data-parallel training, the model-parallel algorithms from Section 2.1 are not redundant, which makes them more vulnerable to hardware and network failures. In fact, even a single failed node can cause permanent loss of trained parameters and leave others unable to run forward and backward passes. To the best of our knowledge, there is only one study about training large models with unreliable devices (Ryabinin & Gusev, 2020), but it supports only Mixture-of-Experts and requires at least 1Gb/s interconnect even in toy experiments.

## 2.3 COMMUNICATION EFFICIENCY & COMPRESSION STRATEGIES

In this section, we discuss techniques that address training with limited network bandwidth and/or high latency, such as gradient compression or overlapping computation with communication phases.

**Efficient parameter gradient communication.** Data-parallel training requires the synchronization of gradients after each backpropagation pass, which can be costly if the model has many parameters or network bandwidth is limited. Deep Gradient Compression (Lin et al., 2018) approaches this problem by sparsifying gradients before synchronizing and corrects the momentum to work with sparse updates. PowerSGD (Vogels et al., 2019) compresses gradients via factorization and uses error feedback to counterbalance approximation errors over time. Techniques like 1-bit SGD (Seide et al., 2014), 1-bit Adam (Tang et al., 2021), and 1-bit LAMB (Li et al., 2021) do not synchronize gradients directly, but instead synchronize error-corrected 1-bit local optimizer statistics. Dettmers (2015) uses nonlinear 8-bit quantization to compress gradients before communication. We use this technique in conjunction with compression-aware architectures. Besides direct compression of gradients, another effective technique is to use layer sharing (Lan et al., 2020) which reduces the amount of parameter gradients that need to be synchronized by the factor of how many times a layer is reused.

**Overlapping communication and computation.** Model, pipeline, and data parallelism all have specific synchronization points at each layer, stage, and parameter update, respectively, and require communication-intensive transfers of gradients or activities. Usually, model training cannot progress until these transfers are completed. One effective way to reduce this cost is to overlap communication with computation and thereby *hiding* the transfer hiding the synchronization latency. This can be achieved by combining multiple parallelization techniques (Krizhevsky, 2014; Rajbhandari et al., 2020), by synchronizing parameter gradients layer-by-layer in lockstep with backpropagation (Paszke et al., 2019), or by using pure pipeline parallelism (Huang et al., 2019; Narayanan et al., 2019). However, pure pipeline parallelism requires many pipeline stages to hide the communication effectively. To overcome this problem, we develop compression-aware architectures that work well even with relatively few pipeline stages. Another approach is to improve the computation/communication ratio instead of overlapping communication. This ratio can be improved by using gradient accumulation (Ott et al., 2018) or large batch sizes (You et al., 2020).

**Model Compression methods.** A seminal study of compression is work by Han et al. (2016) which uses a pipeline of pruning, quantization, and Huffman coding to compress neural networks for inference. Courbariaux et al. (2014) explores trade-offs between the numeric precision during training and the final test error. Work on 1-bit quantization seeks to trade predictive performance for speedups (Rastegari et al., 2016; Courbariaux et al., 2015). While extreme compression methods have been used for smaller models and convolutional networks successfully (Qin et al., 2020), quantization of large models or Transformers has been more challenging (Ramesh et al., 2021).

## 3 COMMUNICATION-EFFICIENT MODEL PARALLELISM

### 3.1 THE SQUARE-CUBE LAW OF DEEP LEARNING

To better understand the general scaling properties of model parallelism, we must abstract away from the application-specific parameters, such as model architecture, minibatch size, and system design. To that end, we first consider a simplified model of pipeline parallelism. Our "pipeline" consists of $k$ stages, each represented by an $n \times n$ matrix. Intuitively, the first stage represents input data and all subsequent stages are applying "layers" to that data. This model does not account for specifics of any single application, we can use it do capture the intuition that will later apply to real-world cases.

During "training", stages iteratively perform matrix multiplication, then send the output to the subsequent pipeline stage over a throughput-limited network. These two operations have different scaling properties. The compute time for naive matrix multiplication scales as $O(n^3)$. While this can be reduced further in theory (Coppersmith & Winograd, 1990; Alman & Williams, 2021), it is only used for very large matrices (Zhang & Gao, 2015; Fatahalian et al., 2004; Huang et al., 2020), the standard for neural networks is to use $O(n^3)$ algorithms.

In turn, the communication phase requires at most $O(n^2)$ time to transfer a batch of activations (or gradients). Therefore, as we increase model size, the computation time grows faster than communication time, regardless of which matrix multiplication algorithm we use. We refer to this idea as the *square-cube law* after the eponymous principle in physics (Galileo, 1638; Allen, 2013).

Figure 1: **(Left)** A simplified example of the square-cube law, **(Right)** Relative device utilization for Transformer layers using Tesla V100 and 500Mb/s network bandwidth (see Section 4.1).

This principle applies to many real-world neural networks architectures, albeit with some confounding variables. In convolutional neural networks (Fukushima, 1980), the computation time scales as $O(BHWC^2)$ and the communication is $O(BHWC)$, where B, H, W and C stand for batch size, height, width and the number of channels. Recurrent neural networks (Rumelhart et al., 1986; Hochreiter & Schmidhuber, 1995) need $O(BLH^2)$ compute in terms of batch size, sequence length and hidden size, respectively, and $O(BLH)$ or $O(BH)$ communication, depending on the architecture. With the same notation, transformers (Vaswani et al., 2017) require $O(BL^2H)$ compute for attention layers, $O(BLH^2)$ compute for feedforward layers, but only $O(BLH)$ communication in both cases.

Based on these observations, we conclude that pipeline parallelism naturally grows more communication-efficient with model size. More precisely, increasing the hidden dimension will reduce the communication load per device per unit of time, making it possible to train the model efficiently *with lower network bandwidth* and *higher latency*[4]. While the exact practical ramifications depend on the use case, Section 4.1 demonstrates some of the larger models trained with pipeline parallelism can already train at peak efficiency with only hundreds of Mb/s bandwidth.

Curiously, the square-cube principle also applies to intra-layer parallelism (Krizhevsky et al., 2012; Jacobs et al., 1991), but using this technique at 500Mb/s only becomes feasible for impractically large layer sizes in excess of 65536 units. Data-parallel training with sharding and/or offloading does not scale as well as its communication time scales the size of *model parameters* instead of activations. However, it may be possible to achieve similar scaling with another type of data-parallel training using low-rank gradient approximations (Vogels et al., 2019).

### 3.2 SWARM PARALLELISM

Traditional pipeline-parallelism can be communication-efficient, but this alone is not enough. Since each swarm participant has different compute and network capabilities, a pipeline formed out of such devices would be bottlenecked by the single weakest link, i.e. the participant with the smallest training throughput. As a result, the more powerful nodes along the pipeline would be underutilized due to either lack of inputs or slow subsequent stages. On top of that, if any single peer fails or leaves training prematurely, it will stall the entire training procedure.

In order to overcome these two challenges, we replace the rigid pipeline structure with randomized temporary "pipelines" formed on the fly during each iteration. Each participant can send their outputs to any peer that serves the next pipeline stage. Thus, if one peer has significantly more compute than others, it can process inputs from multiple predecessors and distribute its outputs across several weaker peers to maximize utilization. Furthermore, if a participant leaves during training, its predecessors can reroute their requests to its neighbors, allowing the training to proceed as long as there is at least one active participant at every pipeline stage (we elaborate on this Appendix C).

The resulting infrastructure consists of multiple consecutive swarms, as depicted in Figure 2. Peers within one swarm serve the same pipeline stage (i.e. the same subset of model layers). During forward pass, peers receive micro-batches of inputs from random predecessors and send activations a random peers in the next stage. For *backward* pass, peers receive gradients w.r.t. outputs and compute gradients w.r.t. layer inputs and accumulates gradients w.r.t. parameters. Each peer has two queues for incoming and outgoing requests to maintain high GPU utilization even with a noticeable communication latency via buffering. Similarly to other pipeline implementations (Huang et al., 2019; Narayanan et al., 2021), SWARM uses activation checkpointing (Griewank & Walther, 2000; Chen et al., 2016) to reduce the memory footprint.

---

[4]Latency slows the communication down by a constant factor that also grows less important with model size.

Figure 2: An overview of SWARM parallelism, illustrating both normal operation, device failures and adaptive rebalancing. One of the stage 2 workers leaves and a worker from stage 3 takes its place.

Once enough gradients are accumulated, peers form groups, run All-Reduce to aggregate gradients within their respective pipeline stages and run a global optimizer step. SWARM can also use Delayed Parameter Updates (DPU) to further improve device utilization by performing the optimizer step in parallel to running the next batch. While this is technically asynchronous, DPU is shown to achieve similar per-iteration convergence as fully synchronous training (Ren et al., 2021; Diskin et al., 2021).

**Stochastic wiring.** To better utilize the capacity of each individual device, we dynamically "wire" each input through each stage and pick devices in proportion to their training throughput. SWARM achieves this by creating several "trainer" processes that route through different pipeline stages using interleaved weighted round-robin algorithm (Katevenis et al., 1991; Tabatabaee et al., 2020) with randomized initial order; these processes do not use GPU and have no trainable parameters. Appendix D contains a detailed description of this component.

**Adaptive swarm rebalancing.** As we described earlier in Section 2.2, our workers can join and leave training at any time. If any single pipeline stage loses too many peers, the remaining ones will have to deal with an increased processing load, which will inevitably form a bottleneck. SWARM parallelism addresses this problem by allowing peers to dynamically switch between pipeline stages to rebalance the training throughput. Every $T=300$ seconds, peers measure the utilization rate of each pipeline stage as the queue size. Peers from the most under-utilized pipeline stage will then switch to the most over-utilized one (see Figure 2), download the latest training state from their new neighbors and continue training. Similarly, if a new peer joins midway through training, it automatically finds the most over-utilized pipeline stage by following the same protocol. As a side-effect, this technique allows SWARM to operate with uneven pipeline stages by allocating peers in proportion to the compute requirements of each pipeline stage.

### 3.3 COMPRESSION-AWARE ARCHITECTURES

Since pipeline parallelism has several distinct points of communication, the network overhead can be reduced considerably by reducing the size of features at these communication points. To exploit this, we develop compression-aware architectures that apply extreme compression at these points. We study two distinct ways of compression: compression through a linear bottleneck layer and compression through a bottleneck induced by the maxout activation function (Goodfellow et al., 2013). We combine our compression-aware architectures with 8-bit quantization and Huffman coding to reduce the communication overhead by another 62%.

**Fully Connected Layers – The Baseline:** Fully connected layers in transformers consist of a multi-layer perceptron with a single hidden layer and a non-linear activation function. Without biases and with a residual connection (He et al., 2015) from the inputs to the outputs this can be described as: $\text{MLP}(\mathbf{x}, \mathbf{w}_1, \mathbf{w}_2) = \sigma(\mathbf{x}\mathbf{w}_1)\mathbf{w}_2 + \mathbf{x}$,

where $\mathbf{x} \in \mathbb{R}^{b \times s \times m}$, $\mathbf{w}_1 \in \mathbb{R}^{m \times h}$, $\mathbf{w}_2 \in \mathbb{R}^{h \times m}$, and $\sigma(\cdot)$, is a non-linear activation function, such as ReLU (Krizhevsky et al., 2012), and $b$, $s$, $m$, and $h$ are the batch, sequence, model, and hidden dimension of the transformer neural network. To compress the output of the MLP layer, we want to apply a compression layer between two stages. For example, if we have 24 layers and 4 stages, we have 3 compression layers at layer 6, 12, and 18 that bridge these stages.

**Bottleneck layers:** We experiment with simple bottleneck layers that work by compressing the output features of the MLP by linear projection:

$$\text{Bottleneck}(\mathbf{x}, \mathbf{w}_1, \mathbf{w}_2, \mathbf{w}_c, \mathbf{w}_d) = \text{LayerNorm}(\text{LayerNorm}(\text{MLP}(\mathbf{x}, \mathbf{w}_1, \mathbf{w}_2))\mathbf{w}_c)\mathbf{w_d}, \quad (1)$$

where $\mathbf{w}_c \in \mathbb{R}^{m \times c}$, $\mathbf{w}_d \in \mathbb{R}^{c \times m}$ are compression and decompression parameters with compression dimension $c < m$. We find it critical to use a layer norm (Ba et al., 2016) to ensure training without divergence. The parameter tensor $\mathbf{w}_c$ resides in one stage and its outputs are transferred to the next stage which holds the parameter $\mathbf{w}_d$, which requires $m/c$ times less communication compared to the original model. Note that adding a bottleneck only adds two linear layers for the forward pass and decreases the size of MLP activations; thus, its computational overhead is negligible.

**Maxout Compression:** Compared to bottleneck compression, maxout compression works by using the maxout activation function (Goodfellow et al., 2013) for compression rather than a linear projection. The maxout function of factor $k$ takes inputs with a hidden dimension of $d$ and reduces this dimension by the factor of $k$ by computing the maximum value for each non-overlapping window of $k$ features. We use maxout compression as follows:

$$\text{Maxout}(\mathbf{x}, \mathbf{w}_1, \mathbf{w}_2, \mathbf{w}_d) = \text{LayerNorm}(\text{maxout}_k(\text{LayerNorm}(\text{MLP}(\mathbf{x}, \mathbf{w}_1, \mathbf{w}_2))))\mathbf{w_d}, \quad (2)$$

where the output is reduced by a factor of $k$ through the maxout function in the previous stage, and these outputs are sent to the next stage which holds the decompression matrix $\mathbf{w}_d \in \mathbb{R}^{m/k \times m}$.

## 4 EXPERIMENTS

### 4.1 COMMUNICATION EFFICIENCY AT SCALE

Before we can meaningfully evaluate SWARM parallelism, we must verify our theoretical observations on communication efficiency. Here we run several controlled experiments that measure the GPU utilization and network usage for different model sizes, using vanilla transformer architecture (Vaswani et al., 2017) with batch size 1 and sequence length 512. To decouple the performance impact from other factors, we run these experiments on homogeneous V100 GPU nodes that serve one pipeline stage over the network with varying latency and bandwidth. The full experimental configuration is deferred to Appendix B.

First, we measure how model size affects the computation to communication ratio at 500 Mb/s network bandwidth in both directions. We consider 4 different model configurations: the base configuration from the BERT paper (Devlin et al., 2019), "xxlarge" ("large" with $d_{model}$=4096), which is used in several recent works (Lan et al., 2020; Sun et al., 2021; He et al., 2020), the full GPT-3 model with $d_{model}$=12288 (Brown et al., 2020) and our modified Transformer architecture from Section 4.3 with $d_{model}$=4096, 3 layers per pipeline stage and 8-bit quantized activations. In the first three configurations, the model consists of 12 Transformer layers placed on 12 servers with a single GPU; in the last one, there are 4 servers, each hosting 3 layers. In addition, we evaluate how the same models perform under different network latency in Table 1.

As depicted in Figure 1 (right), larger models can indeed achieve a better GPU utilization rate, since their communication load grows slower than computation. More importantly, even at a modest 500Mb/s network, the resulting GPU idle time can be pushed into the 10–20% range: either naturally for GPT-3-sized models, or with smaller models through activation compression. Furthermore, these models maintain most of their training efficiency at 100ms latency (Figure 3), which is roughly equivalent to training on different continents (Verizon, 2021).

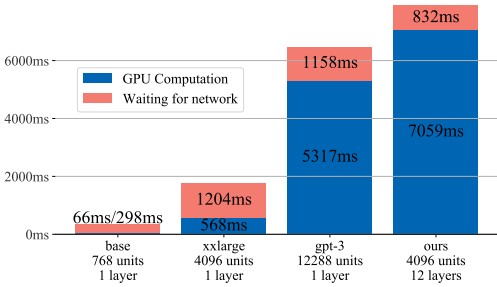

Figure 3: Pipeline computation and idle time per batch at 500Mb/s bandwidth.

Table 1: Relative device utilization at 500Mb/s using simulated network latency.

| Latency (RTT) | Relative GPU utilization, (100% - idle time) | | | |
|---|---|---|---|---|
| | base | xxlarge | gpt-3 | ours |
| none | 18.0% | 32.1% | 82.1% | 89.5% |
| 10ms | 11.8% | 28.9% | 79.3% | 87.2% |
| 50ms | 4.88% | 20.1% | 70.3% | 79.5% |
| 100ms | 2.78% | 14.9% | 60.2% | 71.5% |
| 200ms | 1.53% | 10.1% | 48.5% | 59.2% |

Table 2: Training of language models on the OpenWebText Corpus (OWT). Our baseline model has 253M parameters and is trained for 8 GPU-days. We apply bottleneck and maxout compression to our baseline in 2 and 4 stages with a compression factor between 2-4x. We can see that maxout outperforms bottleneck compression if many stages or high compression factors are used. WT=WikiText, PTB=Penn Treebank, 1BW=Billion word corpus.

| Model | Stages | Compression | Validation perplexity | | | | | |
|-------|--------|-------------|------|---------|------|-------|------|------|
| | | | OWT | LAMBADA | WT2 | WT103 | PTB | 1BW |
| Baseline | – | – | 19.7 | 86.4 | 56.2 | 35.4 | 133.0 | 80.9 |
| Bottleneck | 2 | 2x | **19.5** | 87.7 | **56.5** | **35.2** | 129.8 | 79.2 |
| Maxout | 2 | 2x | 19.6 | **85.4** | 56.6 | **35.2** | 126.8 | **78.8** |
| Bottleneck | 4 | 2x | 21.7 | 100.0 | 66.4 | 40.0 | 149.6 | 89.5 |
| Maxout | 4 | 2x | **21.4** | **89.9** | **63.9** | **39.5** | **142.1** | **86.2** |
| Bottleneck | 2 | 4x | 21.6 | 99.8 | 64.8 | 39.6 | 145.6 | 88.3 |
| Maxout | 2 | 4x | **20.5** | **89.6** | **60.0** | **37.1** | **141.7** | **83.5** |
| Bottleneck | 4 | 4x | 28.9 | 141.6 | 100.2 | 58.1 | 235.5 | 118.3 |
| Maxout | 4 | 4x | **21.3** | **93.5** | **63.6** | **39.2** | **147.7** | **89.1** |

## 4.2 COMPRESSION-AWARE ARCHITECTURES

**Experimental Setup:** As a baseline we train a transformer language model (Vaswani et al., 2017) on the OpenWebText corpus (Gokaslan & Cohen, 2019). We use the following transformer model: sequence size 512, 16 layers with model dimension 1024, and hidden dimension 4096 for a total of 253M parameters. We use BPE encoding (Radford et al., 2019) with vocabulary size of 50264 symbols. We do not use dropout or other regularization since our models underfit. We run our models in fairseq (Ott et al., 2019).

We test bottleneck and maxout compression for a compression factor of 50% and 75% compression compared to the original size over two and four stages. We look at how these compression aware architectures degrade performance compared to the compression that they achieve.

**Results:** The results of our compression aware architectures are shown in Table 2. We can see that while the bottleneck architecture is competitive with maxout for a compression factor of 2x with two stages, maxout has better perplexities if more stages or a higher compression ratio is used. The out-of-distribution perplexities vary consistently with the in-distribution perplexity, which suggest compression-aware architectures do not degrade out-of-distribution performance more than in-distribution performance. As such, maxout compression is an effective technique to reduce the bandwidth requirements of pipeline parallel training further.

Together with the square-cube law for distributed training, and SWARM parallelism, compression-aware architectures allow for better scaling of large neural networks trained over preemptible low network bandwidth peers. Thus, compression-aware architectures improve accessibility and affordability to train large neural networks outside of HPC environments.

## 4.3 REAL-WORLD CONVERGENCE AND SCALING

Finally, to verify the efficiency of SWARM parallelism in a practical scenario, we conduct a series of large-scale distributed experiments using preemptible cloud T4 and A100 GPUs over the non-specialized network. In all experiments below, we train a Transformer language model similar to prior work (Brown et al., 2020; Wang & Komatsuzaki, 2021; Black et al., 2021) with 1.08 billion parameters in total: because of layer sharing, it is equivalent to a 12 times larger network in terms of compute requirements. The detailed setup of all experiments is given in Appendix B.

Specifically, our model consists of 4 stages, each containing a single Transformer decoder block with $d_{model} = 4096$ and 12 layers per pipeline stage, which decreases the size of data that needs to be sent during the aggregation step. We also use 8-bit compression (Dettmers, 2015) for activations and gradients to reduce the communication intensity of training. One additional advantage of using the activation compression (compared to other evaluated approaches) is that it does not require to modify the architecture and evaluate its performance; therefore, we hope to demonstrate that can be a simple yet practical solution for bandwidth-constrained environments.

First, to verify that model parallelism with asynchronous updates does not cause any significant convergence issues, we train the model on the Pile (Gao et al., 2020) dataset with 400 T4 cloud instances, each hosting a single accelerator. As a baseline, we use regular data-parallel training with offloading on 128 A100 GPUs. We report the per-iteration convergence of two setups to amortize the computational performance difference between the setups. Figure 4 shows the results of this experiment: it can be seen that the training dynamics of two approaches are indeed similar, which demonstrates the viability of SWARM parallelism for heterogeneous and poorly-connected devices. We also use the T4 node preemption data of this run to demonstrate the necessity of adaptive rebalancing in a pipeline of unreliable devices; refer to Appendix G for the description.

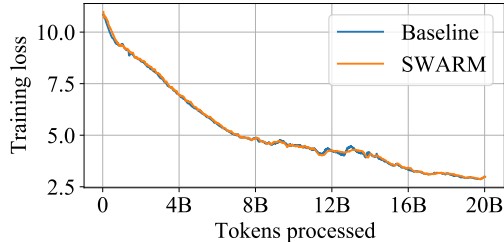

Figure 4: Training convergence comparison.

Table 3: Pipeline throughput comparison.

| Hardware setup | Throughput, samples/s | | Optimal bandwidth, Mb/s | |
| --- | --- | --- | --- | --- |
| | Actual | Best-case | Upload | Download |
| T4 | 17.6 | 19.2 | 317.8 | 397.9 |
| A100 | 16.9 | 25.5 | 436.1 | 545.1 |
| T4 & A100 | 27.3 | — | — | — |

In the next experiment, we aim to measure the pipeline throughput in different hardware conditions and to compare it with theoretical best-case performance. For that, we use several setups: first, we use the same 400 preemptible T4 nodes; also, we use 7 instances with 8 A100 GPU each; finally, we combine these fleets to create a heterogeneous setup. We measure the number of samples processed by the pipeline both in our infrastructure and the ideal use case which ignores all network-related operations. As demonstrated in the left two columns of Table 3, asynchronous training of compute-intensive models with compressed activations allows us to achieve high performance without a dedicated networking solution. Furthermore, the load balancing algorithm of SWARM allows us to dynamically and efficiently utilize different hardware without being bottlenecked by slower devices.

Finally, we use the same load testing scenario to estimate the bandwidth required to fully utilize each device type in the above infrastructure. For this, we measure the average incoming and outgoing bandwidth on the nodes that serve the intermediate stage of the pipeline. We summarize our findings in the right two columns of Table 3: it turns out that with layer sharing and 8-bit compression, medium-performance GPUs (such as T4) can be saturated even with moderate network speeds. Although training over the Internet with more efficient hardware might indeed underutilize the accelerator, this issue can be offset by advanced compression strategies such as compression-aware architectures or more layer sharing iterations.

## 5 CONCLUSION

In this work, we have have analyzed and evaluated the feasibility of high-throughput training of neural models on unreliable, preemptible peers with low network bandwidth. Through analysis guided by the Square-Cube Law of distributed training, we found that this is feasible by training very large models with pipeline parallelism. Specifically, we propose SWARM parallelism to overcome the challenges of pipeline parallelism for preemptible devices with heterogeneous network bandwidths and computational throughputs. We have shown that SWARM parallelism is highly effective at re-balancing peers and maximizing the aggregate throughput of the entire model-parallel pipeline. We showed that the combination of SWARM parallelism and compression-aware architectures allows to train very large models on cheap preemptible cloud instances or on volunteered resources. As such, our work makes the training of large models accessible and affordable to small research teams that do not have access to dedicated infrastructure such as supercomputers.

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

SUPPLEMENTARY MATERIAL

## A    ADDITIONAL RELATED WORK

In this section, we review two additional types of distributed training algorithms that can support training large models.

**Dynamic parameter loading.**    Several recent studies propose alternative execution algorithms that allow training large models with data parallelism. Since neural networks typically use a small fraction of weights at any given moment, the remaining "inactive" weights can be sharded (Rajbhandari et al., 2020) or offloaded to external memory (Pudipeddi et al., 2020; Ren et al., 2021; Rajbhandari et al., 2021). In sharded data-parallelism (Rajbhandari et al., 2020) inactive tensors are distributed across all devices such that each device stores $1/n$th of all parameters. For active layers, the shards are gathered such that each device holds the entire tensor just-in-time for computation. After the computation, the parameters' memory is freed such that only the sharded memory remains (1/Nth per device). This makes it very memory efficient to store model and optimizer states for inactive layers if many devices are available. Similarly to tensor parallelism, these algorithms can support arbitrary models without the need for layer partitioning, and can, in principle, run a large model on a single GPU, which is useful for fine-tuning and inference.

**Architecture-specific methods.**    Finally, some distributed training algorithms take advantage of specific layers, such as locally-connected layers (Dean et al., 2012; Coates et al., 2013), Mixture-of-Experts (Jacobs et al., 1991; Shazeer et al., 2017; Lepikhin et al., 2020), Switch layers (Fedus et al., 2021) or Product Key Memory (Lample et al., 2019). These layers contain many near-independent parts that can be assigned to different devices. They can easily scale to an extremely large number of parameters with a relatively small increase in compute (Shazeer et al., 2017). However, they are also less parameter-efficient (Fedus et al., 2021) and may not apply to all architectures.

## B    DETAILED EXPERIMENTAL SETUPS

### B.1    ADDITIONAL DETAILS FOR SECTION 4.1

We benchmark three popular versions of the Transformer layer:

- Transformer-base: $d_{model} = 768$, $d_{\text{FFN}} = 3072$, 12 heads;
- Transformer-xxlarge: $d_{model} = 4096$, $d_{\text{FFN}} = 16384$, 32 heads;
- "the GPT-3" (Brown et al., 2020): $d_{model} = 12288$, $d_{\text{FFN}} = 49152$, 96 heads.

For simplicity, we set up each experiment with 12 Transformer layers using 12 servers with a single V100-PCIE GPU each. The servers communicate at 500Mbps under 3–6ms latency. Finally, we evaluate GPT-3 with colocated layers — the same architecture as in GPT-3, but this time we place 3 layers per server (4 servers total).

Due to modest communication bandwidth, smaller models spend most of the time waiting for the network. However, that same bandwidth allows for $> 80\%$ GPU utilization when dealing with GPT-3-sized layers and over $90\%$ with GPT-3 layers per device.

The training time reported above is the time required to run forward and backward pass for all layers with a batch of 4×128 tokens, not including the Adam updates. All results are averaged over 1000 consecutive batches, the standard deviations are below 0.1%. All four GPUs are in the same data center, but different servers. Each layer is a TransformerEncoderLayer (torch==1.7.0), wrapped with torch.utils.checkpoint. We use hivemind==0.8.15 (Ryabinin & Gusev, 2020) with a single synchronous trainer based on huggingface BERT training code. However, these results are not specific to hivemind and are likely reproducible in FairScale (Baines et al., 2021) or PyTorch RPC. The only important detail is that the training code should run as much communication as possible in the background, while the GPUs are busy processing batches.

It is important to reuse the same connection for multiple RPC calls, so the TCP buffer does not have to warm up again during each call. Also, our implementation performs quantization asynchronously with communication and other computations.

## B.2 ADDITIONAL DETAILS FOR SECTION 4.3

For the language model training experiment, we use the standard Transformer architecture with two modifications. First, we use Rotary Positional Embeddings (Su et al., 2021) to increase the stability of our training runs. Second, we divide the model into four stages: two stages in the middle consist of regular stacked Transformer layers, the first stage also contains the embedding layer and the last stage includes the language modeling head. To reduce the communication intensity of All-Reduce during the data-parallel steps, we leverage weight sharing (similarly to ALBERT from Lan et al., 2020) in each stage.

## C ANSWERS TO COMMON QUESTIONS

In this section, we provide answers to several assorted questions about our study and address some of the limitations of SWARM parallelism.

**Why not just use data parallelism with offloading?** Regular data parallelism requires all-reduce steps where peers exchange gradients. This can be prohibitively expensive for large models. For example, a 1B parameter model with 16-bit gradients requires 2 GB of data to be synchronized between all $n$ devices. We need at least $n$ messages to perform this synchronization. If we have, for example, 100 devices with bidirectional communication, each client would need to send 2 GB of data to finish the synchronization. Thus, with slow interconnects such synchronizations are not practical.

**Why not just use fully sharded data parallelism with elasticity?** Sharded data parallelism requires all-to-all communication of parameter buffers at each layer. Each of these all-to-all communications can be in parallel and has a size of parameters/$n$ and in total $n$ messages are required. Thus for 1B parameters in 16-bit precision, a total of 2GB of data need to be synchronized for both the forward and backward pass. If using low-bandwidth devices with 100 Mbit/s speed, this would entail an overhead of 5.5 minutes per forward/backward pass which is difficult to hide under computation. This is exacerbated further because all-to-all communication latency is determined by the lowest peer. Thus sharded data parallelism can be particularly ineffective for environments where peers have heterogeneous network bandwidths.

**Zero-offload allows one to train 13B parameters on a single V100, so why do I need SWARM?** Training with zero-offload can slow down training due to the slow bandwidth between external memory and the accelerator. Training with SWARM can *accelerate* training while also allowing the training of large models. Also, see Appendix E for a detailed comparison.

**Should I use SWARM in a supercomputer?** By default, SWARM is worse than traditional parallelism due to extra complexity (see experiments in Section F. However, supercomputers often have heterogeneous devices, in which case SWARM is useful.

**When should I avoid using SWARM?** SWARM is very efficient at training large models with more than 1B parameters. For smaller models a sharded data-parallel approach can be effective. For HPC environments with homogeneous networking, standard sharded data parallel or pipeline parallel training will be more effective than SWARM because the environment is stable and predictable such that re-balancing is not required. For HPC environments which are so extensive that failure of a nodes is likely, the effectiveness of swarm depends on how many nodes are expected to fail. Elastic sharded data parallelism is better than SWARM if the number of expected failures is relatively low.

**How much failure can SWARM handle?** As long as there is at least one operational peer at every pipeline stage and at least one trainer SWARM will work without any issues. The same is true for network partitioning which requires at least one trainer and peer in each pipeline stage.

**Can I use SWARM without layer sharing or quantization?** Yes, SWARM can still be effective in this scenarios. Our bandwidth experiments in the main paper give some idea what the network overhead is. By using no quantization, that means using regular 16-bit activations, the network overhead increases roughly by a factor of two. Without layer sharing the overhead within each

pipeline stage to synchronize the gradient is increased by the number of layers not being shared. As such, a rough estimate of the efficiency of SWARM in these scenarios can be estimated by taking our model-size-network-bandwidth-requirements data and multiplying it by the relevant factor.

**Do the compression-aware architecture modifications apply only to transformers?** Bottleneck and maxout compression are general compression techniques that can be applied to any layer in any architecture. However, their effectiveness may vary depending on where in the model they are applied and what kind of model these are applied to (CNN vs RNN vs Transformer, etc.)

**Some configurations in Section 4.1 measure less than** $20\%$ **GPU idle time, while many HPC systems only achieve** $\approx 80\%$ **GPU utilization. Does this mean that SWARM is** $30\%$ **faster?** No, because these are different measurement types. Narayanan et al. (2021) measures GPU utilization as a fraction of theoretical peak flop/s of their GPUs. In contrast, we only measure what fraction of time is GPU running the model, regardless of efficiency. Since no deep learning workload can achieve $100\%$ peak flop/s, $20\%$ GPU idle time for SWARM means that it can achieve $\approx 0.8\times$ the training throughput compared to training with a infinitely fast network. As a rule of thumb, one can say that SWARM will run at $20\%$ slower than Narayanan et al. (2021) using the infrastructure that is several times cheaper.

## D   STOCHASTIC WIRING DETAILS

Our approach uses *stochastic wiring*, a specialized routing algorithm designed around heterogeneous unreliable devices and high network latency. The core idea of stochastic wiring is to route each training microbatch through randomly chosen pipeline stages such that the workload of each peer is proportional to its performance.

From a system design perspective, each device runs a separate *trainer* process that forms microbatches and routes them through pipeline stages (forward and backward pass). As we describe earlier in Section 3.2, trainers run Interleaved Weighted Round Robin (Katevenis et al., 1991; Tabatabaee et al., 2020) scheduling to dynamically assign microbatches to peers based on each peer's training throughput ("samples per second") in a balanced way.

To achieve this, trainers maintain a moving average estimate of each peer's training throughput based on how long does the peer take to process the batch. This moving average estimate is updated each time when the corresponding peer responds to a request. If a peer does not respond or otherwise fails to process the minibatch, trainer will temporarily ban this peer and reset its throughput estimate.

Curiously, different trainers can have different throughput estimates based on the network topology. For instance, if peers are located in two cloud regions, a given peer's trainer will have higher throughput estimate for peers in the same data center. In other words, trainers automatically adjust to the network topology by routing more traffic to peers that are "nearby".

Another important observation is that *stochastic wiring allows SWARM to mitigate network latency*. Unlike existing pipeline algorithms (Huang et al., 2019), SWARM workers do not get blocked if their neighbors take too long to process a minibatch. Instead, each SWARM device maintains a queue of microbatches assigned by trainers. In case of a latency spike, workers keep processing previously queued microbatches, maintaining high device utilization.

## E   ON THE RELATION BETWEEN SWARM AND ZERO-OFFLOAD

In this section, we argue that depending on the use of DPU, SWARM-parallel training is equivalent to either fully synchronous training or ZeRO-Offload (Ren et al., 2021). That is, SWARM produces exactly the same stepwise updates as conventional distributed training algorithms and will therefore achieve a solution in the same number of steps.

This observation is similar to how many advanced distributed training techniques (Huang et al., 2019; Rajbhandari et al., 2020) are computationally equivalent to regular synchronous training on a single device. For instance, despite using advanced distributed computation strategies, GPipe (Huang et al., 2019) computes exactly the same mathematical expression to obtain gradients and applies those gradients in the same order as any other *synchronous* training algorithm. On the other hand,

PipeDream (Narayanan et al., 2019) changes the order in which the updates are applied, introducing the so-called stale gradients (Recht et al., 2011). This allows PipeDream to improve device utilization, but it has been shown to reduce the final model quality in some setups (Wang et al., 2020).

Despite using randomized routing and asynchronous communication between pipeline stages, SWARM still performs optimizer steps synchronously after peers collectively accumulate a global batch size (hyperparameter). While different peers may accumulate a different number of samples, they will all use the same gradient after averaging. Finally, any peer that fails or does not meet this condition for any reason is considered a straggler and must reload its state from neighbors before it can resume training.

This ensures that all surviving peers use non-stale aggregated gradient over the specified batch size when they perform the optimizer step. The only deviation from fully synchronous training is that SWARM uses the same approach for CPU offloading as ZeRO-Offload, and by extension, delayed parameter updates (DPU). While DPU were shown not to affect convergence (Ren et al., 2021), one can disable this functionality and make SWARM fully equivalent to normal training.

Naturally, these guarantees come at the cost of reduced compute utilization as a small portion of devices will need to wait after every step. However, as we show in Section 4.3, SWARM can still train with competitive training throughput due to the fact that large models are trained with increased batch sizes (Brown et al., 2020).

## F    DETAILED PERFORMANCE COMPARISON

Here, we investigate how SWARM parallelism compares to existing systems for training large models: **GPipe** (Huang et al., 2019) and **ZeRO-Offload** (Ren et al., 2021). The purpose of this section is to compare the training throughput in "ideal" conditions (with homogeneous reliable devices and balanced layers), as deviating from these conditions makes it *infeasible* to train with baseline systems.

We evaluate training performance for sequences of 4 Transformer layers of identical size distributed over 16 workers. The pipeline does not contain embeddings or language modeling heads, as it would result in imbalance between the stages. Similarly to Section 4.1, we use two layer configurations: "xxlarge" ($d_{model}$=4096, $d_{FFN}$=16384, 32 heads) and "GPT-3" ($d_{model}$=12288, $d_{FFN}$=49152, 96 heads). The microbatch size is 4 for "xxlarge" and 1 for "GPT-3", and the sequence length is 512.

To provide a more detailed view of the training performance, we measure two separate performance statistics: the training throughput and the All-Reduce time. The training throughput measures the rate at which the system can process training sequences, i.e. run forward and backward passes. In turn, the All-Reduce time is the time each system spends to aggregate those accumulated gradients across devices. The total time per step can be computed as `batch_size / throughput + all_reduce_time`. Intuitively, training with small batch sizes is more sensitive to the All-Reduce time since the algorithm needs to run All-Reduce more frequently and vice versa.

**Hardware setup:** Each worker uses a V100-PCIe GPU with 16 CPU threads (E5 v5-2660v4) and 128 GB RAM. The only exception is for ZeRO-Offload with "GPT-3" layers, where we had to double the RAM size because the system required 190GB at peak. Similarly to Section 4.1, each worker can communicate at a 500Mb/s bandwidth for both upload and download for a total of 1Gb/s. In terms of latency, we consider two setups: with **no latency**, where workers communicate normally within the same rack, and with **latency**, where we inject additional $100 \pm 50$ms latency in the kernel[5].

**GPipe configuration:** We use a popular PyTorch-based implementation of GPipe[6]. The model is partitioned into 4 stages repeated over 4 model-parallel groups. To fit into the GPU memory for the "GPT-3" configuration, we offload the optimizer into RAM using ZeRO-Offload. Before averaging, we use PyTorch's built-in All-Reduce to aggregate gradients.

**ZeRO-Offload configuration:** Each worker runs the entire model individually, then aggregates gradients with peers. For "xxlarge", we use the official implementation from Ren et al. (2021). However, for "GPT-3", we found that offloading the optimizer still does not allow us to fit 4 layers into the GPU memory. For this reason, we also offload parameters using "offload_param"[7].

---

[5]More specifically, `tc qdisc add dev <...> root netem delay 100ms 50ms`

[6]The source code is available at `https://github.com/kakaobrain/torchgpipe`

[7]Based on `https://www.deepspeed.ai/docs/config-json/#parameter-offloading`

Table 4: Training performance for "GPT-3".

| System | Throughput, samples / s | | All-Reduce time seconds / round | |
|---|---|---|---|---|
| | No latency | Latency | No latency | Latency |
| SWARM | 0.619 | **0.558** | 441.7 | **455.4** |
| GPipe | **0.633** | 0.477 | **403** | 469.6 |
| Offload | 0.382 | 0.382 | 1527.9 | 1635.4 |

Table 5: Training performance for "xxlarge".

| System | Throughput, samples / s | | All-Reduce time seconds / round | |
|---|---|---|---|---|
| | No latency | Latency | No latency | Latency |
| SWARM | 2.358 | 2.161 | 45.36 | **51.269** |
| GPipe | 2.541 | 0.957 | **44.17** | 64.828 |
| Offload | **3.08** | **3.08** | 168.71 | 252.26 |

We report our measurements in Tables 4 and 5. Table 4 demonstrates that for larger model size, SWARM and GPipe have approximately the same performance without latency. When training with latency, SWARM significantly outperforms GPipe, which is likely caused by the asynchronous queueing (see Appendix D). In turn, when training smaller models (Table 5), ZeRO-Offload outperforms both SWARM and GPipe. This result coincides with our earlier observations in Figure 1, where the same model spent the most of the time waiting for communication between pipeline stages.We also observe that ZeRO-Offload takes longer to aggregate gradients, likely because each peer must aggregate the entire model, whereas in SWARM and GPipe peers aggregate a single pipeline stage. The variation between All-Reduce time in GPipe and SWARM is due to implementation differences. Overall, SWARM is competitive to HPC baselines even in idealized homogeneous environment.

## G ADAPTIVE REBALANCING EVALUATION

In this experiment, we evaluate the efficiency of adaptive peer rebalancing between stages proposed in Section 3.2. We use actual statistics of the number of active T4 nodes from the 32-hour segment of the experiment described in Section 4.3 for a representative sample of training dynamics with unstable participation. We use these data to simulate training dynamics as follow: we use a sequence of events, each consisting of a timestamp and the change in the number of peers (which can be positive or negative). When a worker is removed from the pipeline, we randomly choose the stage it was removed from: that is, removing $N$ peers corresponds to $N$ samples from the uniform distribution over four pipeline stages. To compare our method with the baseline without rebalancing, we run 10 simulations over different random seeds and average the resulting trajectories.

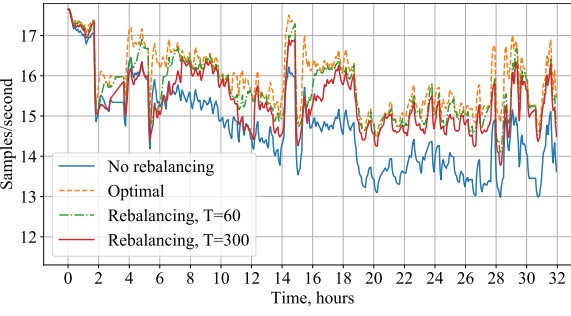

Figure 5: Throughput of rebalancing methods over time.

Table 6: Relative throughput comparison of pipeline rebalancing methods.

| Rebalancing | % of optimal | | |
|---|---|---|---|
| | Overall | First 1h | Last 1h |
| None | 82.7 | 99.0 | 45.4 |
| $T = 300$ | 95.8 | 99.4 | 88.9 |
| $T = 60$ | 97.6 | 99.8 | 91.7 |

The results of this evaluation are available in Figure 5; for reference, we also provide the performance of a theoretically optimal rebalancing strategy that maintains the highest possible throughput at every moment. It can be seen that even with rebalancing period $T = 300$, adding dynamic rebalancing helps significantly improve the overall throughput of the pipeline during the experiment. When the total number of peers is approximately stable, the rebalanced pipeline also reaches the optimal one in terms of throughput, which shows the efficiency of our strategy even when moving only one node at a time. In addition, we observed that for some brief periods, the performance of the unbalanced pipeline exceeded the throughput of the balanced one due to random choice of disconnecting peers (dropping more from "overrepresented" stages affects the imbalanced pipeline less). However, this held true only for $\approx 4.5\%$ of the entire experiment and was quickly mitigated by adaptive rebalancing.

As expected, decreasing $T$ from 300 to 60 seconds improves both the overall throughput and the speed of convergence to optimal pipeline performance; however, the effect is not as drastic compared to the increase in DHT data transfer. This is also demonstrated by Table 6, which shows the relative throughput of the three configurations compared to the optimal one. Furthermore, the table displays that although initially there is little difference between rebalancing choices, it becomes more pronounced later on as the imbalanced version drifts further from the optimal state.

## H  TIME TO SOLUTION

In this section, we evaluate the compression-aware techniques proposed in Section 3.3 from a practitioner's point of view. A natural way to compare these techniques is in terms of "time to solution", i.e. the wall time it takes to achieve the desired validation objective. In practice, this time depends on three main factors: the compression strategy, the distributed training algorithm and the compute infrastructure.

In order to disentangle these factors, we first address the relation between the training algorithm and the compute infrastructure. As we discuss in Section 3.2 (and later in Appendix E), SWARM parallelism has the same per-iteration behavior as other synchronous methods. Theoretically, the choice of an optimal training system should come down to whichever algorithm has the highest training throughput.

To verify this argument in practice, we compare the per-iteration and per-hour performance of SWARM against fully synchronous training. For this experiment, we train the ALBERT model (Lan et al., 2020) on the WikiText-103 dataset (Merity et al., 2017). We use the ALBERT-Large variant with 4 layer groups that correspond to 4 SWARM stages *without the architecture modifications from Section 3.3*. We follow the exact hyperparameters from the original paper, e.g. we use the LAMB optimizer You et al. (2020) with batch size 4096 and sequence length 512. We train this model in three setups: traditional distributed training with 8 V100 workers, SWARM with 8 preemptible V100 GPUs, and SWARM with 32 preemptible T4 workers.

To quantify the time to solution, we measure the wall time required to achieve ALBERT objective equal to **1.5**. Additionally, we report the per-hour cost of each experimental setup and the total cost of achieving a loss of 1.5 using public cloud provider pricing estimates.

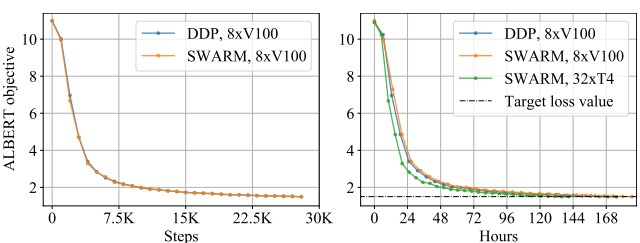

Figure 6: Convergence curves of ALBERT with SWARM and standard data-parallel training.

Table 7: Training time and costs.

| Setup | Time, hrs | Cost, $ | |
| --- | --- | --- | --- |
| | | Hourly | Total |
| $8 \times V100$ reliable | 175.4 | 7.834 | 1374 |
| $8 \times V100$ preemptible | 192.6 | 5.383 | 1037 |
| $32 \times T4$ preemptible | 140.8 | 3.536 | 497.8 |

Figure 6 (left) demonstrates that SWARM matches the per-iteration learning curves of traditional distributed training (PyTorch DistributedDataParallel) up to the random seed. However, SWARM parallelism can achieve $Loss{=}1.5$ more cost-efficiently and/or faster by using preemptible instances. In turn, *when forced to use homogeneous and reliable GPUs*, SWARM would have slightly inferior performance compared to conventional algorithms, which is expected (see Appendix F).

**Compression-aware architectures.** Once we establish the baseline performance, we can analyze how different compression-aware bottlenecks from Section 3.3 affect the time-to-solution tradeoff.

For this experiment, we train a Transformer language model with adaptive inputs (Baevski & Auli, 2019) on the WikiText-103 dataset and measure how compression-aware architecture variants affect convergence. More specifically, we measure the number of iterations it takes to converge to the training perplexity of **22**, which is slightly below the perplexity of GPT2-Large Radford et al. (2019) on this dataset. We evaluate the baseline model and the three compression-aware modifications from Section 3.3: Bottleneck, Maxout, and simple 8-bit quantization, each with 2 pipeline stages and each a compression factor of 2x.

Table 8: Performance of compression techniques for transformer language model with adaptive inputs on WikiText-103. The asterisk denotes that the difference is not statistically significant.

| Compression | Steps to ppl 22 | Communication | Extra compute (ms) | Extra compute (relative) |
|---|---|---|---|---|
| None (baseline) | 1x | 1x | 0 | None |
| Quantization | 0.97x* | 0.5x | 1.2ms | None (overlapped) |
| Bottleneck | 1.26x | 0.5x | 1.96ms | $\leq 0.1\%$ |
| Maxout | 1.28x | 0.5x | 2.04ms | $\leq 0.1\%$ |

In Table 8, we report the relative number of steps it takes to achieve the target perplexity for each architecture variant. We also report the communication rate and additional computation time for these compression methods. Using these results, one can determine which method is optimal for their hardware setup. For instance, training with Maxout with 2 pipeline stages needs 28% more steps, but accelerates the communication phase by $2\times$. Overall, both Maxout and bottleneck layers reduce communication faster than they increase the number of steps. With 1Gb/s or slower interconnect, both compression techniques reduce the time until convergence. However, the same two techniques would result in slower training in an ultra-high-bandwidth setup, where the communication speed is not a limiting factor.

In turn, 8-bit quantization reduces communication cost without slowing down per-iteration convergence, making it a "safe bet" for situations where the per-iteration convergence must be preserved. This observation is further supported by our main experiment in Figure 4, where a larger model with 8-bit quantization achieves the same per-iteration convergence as the baseline. In turn, if a higher number of steps is not an issue, one can combine quantization with Bottleneck or Maxout to achieve even more significant reductions in the data transfer volume.

# I   ADDITIONAL SCALING EVALUATION

In this experiment, we investigate the influence of the number of nodes running SWARM-parallel training on the throughput of the pipeline. Specifically, we measure the performance of training the same model as in Section 4.3 in several configurations that differ in the size of the data-parallel group at each pipeline stage, with the number of single-GPU instances ranging from 8 to 128 (the highest quantity of preemptible nodes that we could reliably maintain for a long time). To isolate the effect of worker heterogeneity, here we use only the T4 accelerators and measure the average performance over 30 minutes of training.

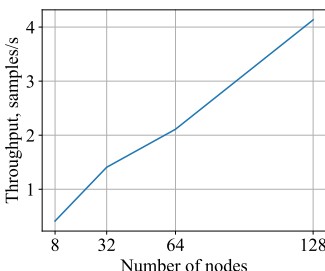

Figure 7: Scaling of SWARM Parallelism throughput with respect to the number of peers.

Figure 7 shows the results of our evaluation. It can be seen that the training performance exhibits an approximately linear scaling pattern, which can be explained by the high algorithmic efficiency of both SWARM stochastic wiring strategy and the auxiliary training components such as the DHT.

