# OpenReview forum: "SWARM Parallelism: Training Large Models Can Be Surprisingly Communication-Efficient"
_ICLR.cc/2022/Conference — ICLR 2022 Submitted_

### Official Review · Reviewer_5Pxy · 2021-11-02

**Correctness:** 3
**Technical Novelty And Significance:** 4
**Empirical Novelty And Significance:** 3
**Recommendation:** 6
**Confidence:** 4

**Details Of Ethics Concerns:**

In the scenario the authors work on, it seems training data has to be distributed to volunteered training machines, which would increase the data leakage risk.

**Main Review:**

Strengths:
1. The literature review of this paper is adequate and great.
2. The proposed SWARM framework is novel and practical. The authors "square-cube law" of communication-computation relation in pipeline parallel is convincing. The description of the framework is pretty clear.
3. The authors verify their idea on both simulated and real scenarios, which makes the result quite solid.


Weakness:
The authors should provide more details about their methods, including:
1. How to choose peers to construct a swarm for each stage? Since the framework is designed for heterogeneous devices, which may of different geographic, computation capacities, interconnect speeds, etc., and within each swarm, all reduce operation is required to aggregate gradients, it would be better to describe your swarm construction method and provide some experimental results to show the advantage of your construction method. Additionally, the swarm division is changeable, is it possible to make the pipeline stage division changeable too?
2. Is gradient compression techniques required during all-reduce?
3. Could you describe "Stochastic wiring" in more detail? Does that mean the device wirings among different stages are fixed during a period of time and modified dynamically after this time slot? If that is true, besides interleaved weighted round-robin algorithm, could you give more details about how to wire these devices, for example, which factors should be taken into account during wiring?
4. Could you give some experimental evaluation about the effect of stochastic wiring and adaptive swarm rebalancing?
5. For the compression techniques comparison, besides end-to-end evaluation on ppl, it would be better to compare their reconstruction error additionally.
6. For experiment part in section 4.1, it would be better to put the configuration "12 servers for 12 transformer layers" in this part instead of in supplementary to help authors understand well. And please keep your description consistent in Figure 3, Table 1 and the main body (BERT-base vs base, BERT-large vs xxlarge).
7. For the results in Table 1, whey the GPU utilization for base model so small even when latency is none? How do you get the conclusion that "these models maintain most of their training efficiency at 100ms latency"?
8. Could you explain the throughput relationship among T4,A100 and T4&A100 ? Why is the T4&A100's throughput much better than others?
9. For real-world scenario, it would be better to provide a visualization figures about dynamic stochastic device wiring and adaptive swarm rebalancing during training.


**Summary Of The Paper:**

This paper proposes a novel model-parallel distributed training framework named Stochastically Wired Adaptively Rebalanced Model Parallelism (SWARM) to improve the training efficiency of large-scale models using unreliable heterogeneous devices and slow interconnect. This method is based on an observation for pipeline-parallelism methods, which need to communicate activations instead of model parameters/ gradients of parameters. The observation is that training larger models with pipeline-parallel will reduce the relative overhead of communication. Combined with compression techniques, the proposed method can train a billion-scale Transformer language model with high training throughput on preemptible low-power T4 GPUs and <400Mb/s network.

**Summary Of The Review:**

This paper proposes a novel and practical method to do pipeline-parallel training for large scale models. The literature review is adequate and experimental evaluation is convincing. Although some details of the system design is missing, I am more than happy to raise my score is they can add those details to make this paper more concrete and useful.

---

> ### Author Response · Authors · 2021-11-11
> **Author Response to Reviewer 5Pxy (Part 2)**
>
> > Could you explain the throughput relationship among T4,A100 and T4&A100 ?
>
> The key observation is located in the paragraph describing the respective experiment: namely, the combined throughput of 400xT4 and 56xA100 *should* be greater than for each separate GPU type, as the raw computational performance is simply added up. However, without load balancing the heterogeneous pipeline will be bottlenecked by the performance of the slowest device (which is T4 in that case): the goal of our experiment was to show that we achieve a reasonable speedup when using different devices across the training infrastructure.
>
> > For real-world scenario, it would be better to provide a visualization figures about dynamic stochastic device wiring and adaptive swarm rebalancing during training.
>
> We thank the reviewer for this idea; we will record the dynamic wiring and rebalancing statistics in our targeted experiments and add them to the appendix (mostly due to the space limits).
>
> ### References
>
> [1] PipeDream: Fast and Efficient Pipeline Parallel DNN Training. Aaron Harlap, Deepak Narayanan, Amar Phanishayee, Vivek Seshadri, Nikhil Devanur, Greg Ganger, Phil Gibbons. 2018
>
> [2] MadPipe: Memory Aware Dynamic Programming Algorithm for Pipelined Model Parallelism. Olivier Beaumont, Lionel Eyraud-Dubois, Alena Shilova. 2020
>
> [3] 8-Bit Approximations for Parallelism in Deep Learning. Tim Dettmers. 2015

---

> > ### Comment · Reviewer_5Pxy · 2021-11-19
> > **Follow up**
> >
> > About the concern of data leakage risk, can you explain the data storage issue when SWARM is used? For example, where should the data stored? Is it on a central storage or distributed to training node?

---

> > > ### Author Response · Authors · 2021-11-21
> > > **Response to the follow-up question on data storage**
> > >
> > > We recognize that data storage is an important aspect of distributed training systems. Depending on the use case, SWARM can rely on some of the existing best practices for dealing with different kinds of data: in short, we stream examples from a central storage, but alternative solutions are also possible.
> > >
> > > The primary use-case of SWARM is pretraining large models, which is usually done on large collections of *public* data, such as Common Crawl [1, 2, 3]. Of course, not all data can be made public. In fact, the existing public datasets usually undergo certain anonymization and/or filtering procedures before they are released [2, 3]. Once the dataset is properly filtered and anonymized, it can be stored either in a centralized database or in a distributed storage such as IPFS, depending on the application.
> > >
> > > The only remaining challenge is an engineering one: how to efficiently provision a large dataset to many workers. One promising solution that we used in Section 4.3 is to stream the data directly from the database, as described in [4]. This approach allows new peers to immediately begin training without having to download the full dataset.
> > >
> > > A more extreme setup is when the dataset cannot be made public, as is the case with Federated Learning. In principle, SWARM can still be applied in this setup with minor modifications. For instance, with cross-silo FL setup, it should be possible to utilize SWARM parallelism with restrictions on stochastic wiring. That is, to form stochastic "pipelines" from peers that have access to the same portion of the dataset, e.g. computers from the same organization. Naturally, this technique should be combined with existing federated learning algorithms for privacy-preserving gradient aggregation [5]. Overall, we believe that this is a promising research direction that deserves a separate study.
> > >
> > > ### References
> > > [1] https://commoncrawl.org/
> > >
> > > [2] The Pile: An 800GB Dataset of Diverse Text for Language Modeling. Leo Gao, Stella Biderman, Sid Black, Laurence Golding, Travis Hoppe, Charles Foster, Jason Phang, Horace He, Anish Thite, Noa Nabeshima, Shawn Presser, Connor Leahy. 2021
> > >
> > > [3] https://github.com/allenai/allennlp/discussions/5056
> > >
> > > [4] Datasets: A Community Library for Natural Language Processing. Quentin Lhoest, Albert Villanova del Moral, Yacine Jernite, Abhishek Thakur, Patrick von Platen, Suraj Patil, Julien Chaumond, Mariama Drame, Julien Plu, Lewis Tunstall, Joe Davison, Mario Šaško, Gunjan Chhablani, Bhavitvya Malik, Simon Brandeis, Teven Le Scao, Victor Sanh, Canwen Xu, Nicolas Patry, Angelina McMillan-Major, Philipp Schmid, Sylvain Gugger, Clément Delangue, Théo Matussière, Lysandre Debut, Stas Bekman, Pierric Cistac, Thibault Goehringer, Victor Mustar, François Lagunas, Alexander M. Rush, Thomas Wolf. 2021
> > >
> > > [5] Practical Secure Aggregation for Federated Learning on User-Held Data. Keith Bonawitz, Vladimir Ivanov, Ben Kreuter, Antonio Marcedone, H. Brendan McMahan, Sarvar Patel, Daniel Ramage, Aaron Segal, Karn Seth. 2016 https://arxiv.org/abs/1611.04482

---

> ### Author Response · Authors · 2021-11-11
> **Author Response to Reviewer 5Pxy (Part 1)**
>
> Thank you for the detailed review and a thorough evaluation of our paper! We address your concerns to the best of our ability below; if you have further questions or need additional clarification, we would be happy to answer them throughout the discussion period.
>
> > How to choose peers to construct a swarm for each stage?
>
> Thank you for this question! Although it is quite difficult to conduct a reproducible experiment in a truly heterogeneous environment, the concern of peer imbalance was precisely what motivated us to develop both the stochastic wiring component (in particular, to use the weighted round-robin algorithm) and to implement adaptive swarm rebalancing. Simply put, the exact balance of peers across swarms at construction time is not necessary, because as the throughput of each pipeline stage is estimated, nodes migrate to the bottlenecked parts to improve overall throughput.
>
> > the swarm division is changeable, is it possible to make the pipeline stage division changeable too
>
> It is indeed possible to optimize layer partitioning in model-parallel training: in fact, there exist several works studying this specific problem (for instance, [1] and [2]). Because the network architecture is usually fixed after the training starts, the only reason why it might be useful to change the partitioning would be to balance the load across peers, which is resolved by regrouping the peers themselves. Thus, we believe that this is indeed an important direction for future work, but it is slightly out of scope for our paper.
>
> > Could you describe "Stochastic wiring" in more detail?
>
> We appreciate your request for clarification and will update the paper with more details shortly. For now, let us answer the questions asked in the review: for each single training process (which exists outside of the pipeline and only routes the activations and gradients between nodes without any computation), there is a queue of peers for each stage of the pipeline, and for each forward pass, peers are iteratively chosen from this queue. If a microbatch is successfully processed by the peer, the trainer updates the queue with its throughput and proceeds to the next pipeline stage; if there is an error (such as a disconnect or timeout), the peer is “banned” from the queue and the trainer picks the next available worker.
>
> The order in the queue is maintained in accordance with the interleaved weighted round-robin algorithm: roughly speaking, we keep the order in such a way that nodes in one swarm are queried with a frequency proportionate to their estimated throughput. The dynamic rebalancing of peers across pipeline stages only concerns the location of nodes in swarms, but not the order in which they are queried by trainer processes.
>
> > Could you give some experimental evaluation about the effect of stochastic wiring and adaptive swarm rebalancing?
>
> Thank you for this suggestion! We will conduct these experiments on a smaller scale (to isolate the effect of our proposed methods from other performance considerations) and report the results by the middle of the next week so that it will be possible for you to examine their results and provide your feedback on them.
>
> > For the compression techniques comparison, besides end-to-end evaluation on ppl, it would be better to compare their reconstruction error additionally.
>
> Reconstruction error of dynamic tree quantization is discussed at length in previous work for different families of distributions [3]: in this work, the average relative error is in the range of 2.2 - 2.4% for a sample of 100 tensors. However, for other methods considered in this work, such as the maxout compression and the linear projection, it is impossible to compute the reconstruction error, as the procedure is not invertible; in other words, the outputs of these operations are not comparable and a reconstruction error measure is not meaningful.
>
> > For experiment part in section 4.1, it would be better to put the configuration "12 servers for 12 transformer layers" in this part
> > keep your description consistent in Figure 3, Table 1 and the main body (BERT-base vs base, BERT-large vs xxlarge)
>
> Thank you for these suggestions on the presentation and clarity of our results! We will incorporate these changes in the next revision during the discussion period so that you will be able to see them and tell us if your concerns were resolved.
>
> > For the results in Table 1, why the GPU utilization for base model so small even when latency is none?
>
> To the best of our knowledge, this is caused by the limited network bandwidth. As we describe in Section 4.1, all latency evaluations were performed under a network bandwidth of 500Mb/s for download and 500Mb/s for upload. Since the base model has a higher communication to computation ratio, the GPU still becomes bottlenecked by the network.

---

### Official Review · Reviewer_8gAi · 2021-11-02

**Correctness:** 3
**Technical Novelty And Significance:** 3
**Empirical Novelty And Significance:** 2
**Recommendation:** 6
**Confidence:** 5

**Main Review:**

Pros:
1. The proposed SWARM algorithm design addresses some key shortcomings of existing algorithms such as fault-tolerance, load-balancing, heterogeneous devices, and low bandwidth.
2. The paper addresses an important problem that can reduce the cost of training very large neural networks and make the training accessible to more researchers.
3. The paper is well-written and a thorough background is given.
4. The code and hyperparameters are detailed in an anonymous GitHub repository including recipes for reproducing the experiments.

Cons:
1. My biggest concern is that there is not a single speedup comparison to existing model-parallel techniques (such as [1]) or ZeRO offloading [2].
2. The additional compute time of adding the bottleneck layers should be presented and justified as it might overshadow the benefits of the compression-aware technique.
3. Table 2 shows that adding more stages increases the perplexity (worse) and therefore the solution is limited to a small number of stages. Furthermore, it is unclear why GPT-2 is not the better choice as it has less than half of the parameters and significantly better perplexity. Finally, the speedup gains of the different compression ratios are missing, and therefore it is unclear the direct impact of the compressions other than increasing the perplexity.
4. The paper claims to overcome high latency, however, it does not provide a solution or present justifying results. Although the compression-aware technique addresses the low bandwidth limitations, it does not hold for high latency.
5. The square-cube law was first presented in the introduction section as “a counterintuitive observation that, for some methods, training larger models can actually decrease the network utilization”. However, the given experimental results clearly show the opposite, larger networks have less idle time. In no place throughout the paper have I seen that the first claim been justified.

Having some speedup experiments and comparison to existing algorithms would help increase the score. There is a misconception about ZeRO in Appendix C: Ring All-Reduce is used in practice for gradient aggregation, and therefore the communications of a single device would be 2GB and not 200GB as stated.

Typos:
1. Page 1 Line #7: “both these” → “both of these”
2. Page 4 Line #13: “in Section 4.1 demonstrates” → “Section 4.1 demonstrates”
3. Page 6 Line 19: “finds the most” → “to the most”
4. Page 6 Line 34: “where” → “, where”
5. Page 7 Line 1: “Where” → “where”

[1] Huang, Y., Cheng, Y., Bapna, A., Firat, O., Chen, D., Chen, M., Lee, H., Ngiam, J., Le, Q.V. and Wu, Y., 2019. Gpipe: Efficient training of giant neural networks using pipeline parallelism. Advances in neural information processing systems, 32, pp.103-112.
[2] Ren, J., Rajbhandari, S., Aminabadi, R.Y., Ruwase, O., Yang, S., Zhang, M., Li, D. and He, Y., 2021. Zero-offload: Democratizing billion-scale model training. arXiv preprint arXiv:2101.06840.

**Summary Of The Paper:**

The paper focuses on the model-parallelism setting which is commonly used for training very large neural networks due to its memory efficiency. Today, model-parallelism is mainly used on dedicated HPC clusters which are not available to most researchers, especially for those in developing countries. Existing cost-effective algorithms can significantly reduce the cost of training a neural network, but they currently do not support the model-parallelism setting. Section 2 provides a well-written background on existing techniques in model-parallelism and communication reduction.

To overcome the limitations of existing algorithms, the paper presents the SWARM algorithm - a decentralized model-parallel algorithm that supports heterogeneous devices and node failure recovery. In SWARM, the devices are dynamically assigned to stages that are load-balanced (adaptive swarm rebalancing) throughout the training. Each device dynamically communicates with a device in the subsequent stage in proportion to their training throughput (stochastic wiring). The forward and backward computations in a stage are computed in parallel across devices. Once enough gradients are accumulated, the devices run an All-Reduce (which can be executed asynchronously in the background) to aggregate gradients within their respective pipeline stages and run a global optimizer step. Finally, a compression-aware architecture modification is presented which reduces the dimension of the edge stage layers. This decreases the communication overhead and is further combined with existing compression algorithms.

The experiments section, which I found to be the weakest point in the paper, evaluates the SWARM algorithm and the proposed Transformer architecture with compression-aware modifications. Section 4.1 presents the idle time of the different neural architecture where the newly proposed Transformer architecture has the least idle time. However, the comparison is executed in different settings and configurations, such as 12 vs 1 layer per pipeline stage and 8-bit activation quantization which is not a novelty for this work. Section 4.2 compares the perplexity on different NLP datasets. The performance of the new architecture degrades as the number of stages is increased and therefore the solution is limited to a small number of stages. Furthermore, the speedup gains of the different compression ratios are missing, and therefore it is unclear the direct impact of the compressions other than increasing the perplexity. Finally, Section 4.3 presents a real-world large-scale experiment with 400 T4 GPUs on the Pile dataset. However, it is unclear the speedup gains of the SWARM algorithm, nor is SWARM compared to any existing model-parallel techniques


**Summary Of The Review:**

I recommend rejecting this paper in its current form. Although the paper is well-written and I liked the proposed algorithm, I found the experimental section to be lacking crucial experiments to justify the paper’s claims. My biggest concern is that there is not a single speedup comparison to existing model-parallel techniques. Furthermore, several key details regarding accuracy drops and computation overhead negate the paper’s empirical impact. Having some speedup experiments and comparison to existing algorithms would help increase the score.


### Update Following Revision #2
Throughout the rebuttal process, most of my questions and concerns were fulfilled by both answers and paper revisions. I'm still concerned about the proposed compression technique, which I find the weakest part of this paper. However, with that, I think this paper in its current form should be accepted. The paper addresses an important topic with a high potential of impact on the ML community, both as a tool and research direction.

---

> ### Author Response · Authors · 2021-11-13
> **Author Response to Reviewer 8gAi (Part 2)**
>
> > The square-cube law was first presented in the introduction section as “a counterintuitive observation that, for some methods, training larger models can actually decrease the network utilization”. However, the given experimental results clearly show the opposite, larger networks have less idle time. In no place throughout the paper have I seen that the first claim been justified.
>
> As we discuss in Section 3.1 and demonstrate in Section 4.1, training larger networks results in less **GPU** idle time. In other words, the compute device is better utilized because it spends less time waiting for the network: see Figure 1 (right), Figure 3, and Table 1. Therefore, larger models can reach the same training effectiveness with smaller network utilization.
>
> The misunderstanding probably comes from the fact that “the network utilization” in the quoted sentence refers not to the neural network, but the hardware network connecting the servers for distributed training. If the reviewer deems it necessary, we can emphasize the difference between the two kinds of utilization mentioned in Section 4.1; however, we note that currently it is already stated that we measure the influence of the model size on the *distributed communication load*.
>
> > Appendix C: Ring All-Reduce is used in practice for gradient aggregation, and therefore the communications of a single device would be 2GB and not 200GB as stated.
> > Typos
>
> Thank you for pointing these out, in particular, the typo in Appendix C which could have confused the readers. We will fix these errors in an upcoming update to the paper.

---

> > ### Comment · Reviewer_8gAi · 2021-11-13
> > **Follow-up #1**
> >
> > Thank you for the detailed response.
> >
> > > Speedup Comparison with ZeRO.
> >
> > I'm looking forward to seeing these results. I'm expecting results of the sorts of actual speedups, such as words per second, total compute time per iteration or total compute time of the training. Furthermore, the comparison should be conducted when the two algorithms are running the exact same neural architecture with the same hyper-parameters that affect execution speed, such as batch size. This would allow a fair comparison between the two while disregarding the heterogeneous unreliable nodes which have many viable solutions both in theory and in practice.
> >
> > > The additional compute time of adding the bottleneck layers.
> >
> > The results you provided are a good start, but still do not answer my question since the WPS is computed with regard to both computations and communications. I'm interested in the actual compute time overhead of adding these layers and not the total effect which of course comes with significant communication reduction.
> >
> > > GPT-2
> >
> > In this case, I would remove the GPT-2 results as this is not a fair comparison. I'm expecting a valid comparison of the different proposed compression techniques and ratios with an existing benchmark such as BERT or Transformer-XL.
> >
> > > Speedup Gains
> >
> > Since the bottleneck layers add computation time and the neural networks are run slightly differently, I don't think we can derive the speedup gains from the given results. In my experience, higher GPU utilization doesn't always translate to faster training times. I'm expecting speedup results of the sort of WPS or iteration train time. Of course, both the neural network and the hyper-parameters that affect the execution time should be consistent across these experiments.
> >
> > > Lower Latency Claims
> >
> > I suggest removing the latency claims from the paper as in the current form it is clearly incorrect.
> >
> > > Square-cubs Law
> >
> > I understand my confusion, thank you for clarifying. I think the "decreased network utilization" can be replaced with "decreased network overhead" which might help solve this confusion.

---

> > > ### Author Response · Authors · 2021-11-25
> > > **Response to Follow-up #1**
> > >
> > > We thank the reviewer for clarifications. To keep our discussion structured, we tag the updates in November 21 revision with **[update 2]** and the subsequent revision as **[update 3]**.
> > >
> > > > Speedup Comparison with ZeRO. I'm expecting results of the sorts of actual speedups, such as words per second, total compute time per iteration or total compute time of the training.
> > >
> > > **[update 2]** We did our best to address this claim in the latest revision of the paper. Appendix F contains a practical speed-up comparison against ZeRO-Offload and GPipe in terms of sequences per second (i.e. tokens per second x 512).
> > >
> > > **[update 3]** In Appendix H, we also explore a related question of how does the training throughput translate to the actual time-to-solution with different hardware setups and different architecture variants. We are willing to discuss these results and provide additional clarifications if requested.
> > >
> > >
> > > > I'm interested in the actual compute time overhead of adding these layers and not the total effect which of course comes with significant communication reduction.
> > >
> > > We agree with the concern. However, the numbers we reported earlier are already computed without accounting for communication. The actual compute cost of the bottlenecks turned out to be insignificant compared to the random variation between runs: this follows from the fact that the total model size is 253M parameters and adding a bottleneck consisting of two 1024x512 linear layers results in $\approx$1M additional parameters, which translates to $<1%$ of additional compute for this combination of architecture and sequence length.
> > >
> > > **[update 3]**
> > > We investigate this further in appendix H (part 2), where we report compute overhead separately. Overall, the small compute overhead can be attributed to two factors. First we deliberately optimized our implementation to reduce the overhead for the costly experiments in Section 4.3. Second, the overhead of compressing activations also scales slower than the total computation.
> > >
> > > > In this case, I would remove the GPT-2 results as this is not a fair comparison. I'm expecting a valid comparison of the different proposed compression techniques and ratios with an existing benchmark such as BERT or Transformer-XL.
> > >
> > > **[update 2]** We removed GPT-2 as the reviewer suggested.
> > >
> > > **[update 3]** We also did our best to provide additional comparison in Appendix H, using ALBERT and a Transformer language model on WikiText-103.
> > >
> > > > I suggest removing the latency claims from the paper as in the current form it is clearly incorrect.
> > >
> > > **[update 2]** We tried to improve our explanations of how SWARM affects network latency, added a detailed discussion of this aspect in Appendix D and included experiments with latency in Appendix F.
> > >
> > > In short, the stochastic wiring procedure and asynchronous communication described in Section 3.2 naturally leads to a behavior that mitigates latency. Each peer maintains a queue of incoming forward/backward requests from trainers and processes these requests in batches. These queues “amortize” the network latency and ensure that peer’s GPU has a backlog of tasks to process.
> > >
> > > > I think the "decreased network utilization" can be replaced with "decreased network overhead"
> > >
> > > **[update 2]** We thank the reviewer for this suggestion and change the wording accordingly.

---

> ### Author Response · Authors · 2021-11-13
> **Author Response to Reviewer 8gAi (Part 1)**
>
> Thank you for your feedback and insightful suggestions on our work! Please allow us to address your concerns below:
>
> > There is not a single speedup comparison to existing model-parallel techniques (such as [1]) or ZeRO offloading [2].
>
> While traditional model parallelism does not support our target training scenario (heterogeneous unreliable nodes), we agree that it would be beneficial to compare the performance in a setup without node failures. We will run such an experiment and update our paper by the middle of the next week. As for offloading, *comparing* with it may be a bit misleading since SWARM already uses this technique on each pipeline stage (see page 6, §1): in other words, offloading is complementary to the model parallelism. However, we agree that comparing with ZeRO offloading would be valuable to many readers and will include a discussion of that in the appendix of the next version of the paper.
>
> > The additional compute time of adding the bottleneck layers should be presented and justified
>
> Thank you for the suggestion! The bottleneck layers in most cases reduce computation time, since the size of intermediate representations ($c$) is smaller than for regular layers ($m$). The additional LayerNorm and Maxout layers have only a small computational overhead.
>
> For reference, we provide the performance of 2 stages and 1 bottleneck layer (WPS=words per second, higher is better):
>
> Baseline: 72800 WPS
>
> Bottleneck 2x: 73000 WPS
>
> Bottleneck 4x: 73200 WPS
>
> Maxout 2x: 72600 WPS
>
> Maxout 4x: 73400 WPS
>
>
> Thus, using the bottleneck layers results in almost no change in the computational runtime performance of the model. We will clarify this aspect of the bottleneck compression in the revised version of the paper.
>
> > It is unclear why GPT-2 is not the better choice as it has less than half of the parameters and significantly better perplexity.
>
> GPT-2 does better because it is trained for longer on a non-public dataset. We use public data and train for less time, which results in a different baseline perplexity. We would reach similar perplexity if we would train for longer, but we do not have the sufficient computational resources to do so for all experiments.
>
> > Finally, the speedup gains of the different compression ratios are missing, and therefore it is unclear the direct impact of the compressions other than increasing the perplexity.
>
> The speedup ratios for communication are proportional to the compression ratio: for example, with 2x compression, there is 2x faster communication. Table 1 shows the proportions of communication (GPU idle time) and computation time for different architectures. The data from Table 1 in conjunction with the compression ratio can be used to compute speedups. For example, for a base model with no latency, communication amounts to 82% of the time spent with 18% for computation. With a compression-aware architecture that has a 2x compression rate, this improves to 69% of communication time and 31% of computation time – a speedup of 1.69x. Other speedups can be calculated in the same manner.
>
> > The paper claims to overcome high latency, however, it does not provide a solution or present justifying results.
>
> Thank you for pointing out this issue! Indeed, the high latency aspect was not explained very clearly in the submitted version, so we plan to elaborate on it in the next revision during the response period.
>
> The key parts of our solution are (1) increasing the computational intensity of pipeline stages, as described in the second paragraph on page 5 and in Table 1, and (2) using queues for nodes hosting all pipeline stages, which allows us to amortize the request latency by processing inputs from several trainer processes and thus maintain high pipeline utilization.

---

### Official Review · Reviewer_RSKK · 2021-11-02

**Correctness:** 2
**Technical Novelty And Significance:** 3
**Empirical Novelty And Significance:** Not applicable
**Recommendation:** 3
**Confidence:** 5

**Main Review:**

Whilst the research question at hand is definitely valuable and of importance, the study presented by the authors presents with several shortcomings.
- The background and related work section lacks detail in the field of model-parallelism, see e.g.
Ben-Nun, T., & Hoefler, T. (2019). Demystifying parallel and distributed deep learning: An in-depth concurrency analysis. ACM Computing Surveys (CSUR), 52(4), 1-43.
On the other hand, the section on data-parallelism is quite extensive even though not really relevant for this paper. Moreover, some of the work in this section is comparably old and not state-of-the-art any more
- The authors formulate a Square-Cube-Law to justify their choice for pipelining as parallelisation strategy. They argue that for neural network layers relying on matrix multiplication (MLP, Attention), the computation of the layer scales with O(n^3) (where n is the number of nodes), while communication between layers only scales with O(n^2), hence for large layers, computation will be the bottleneck. However, this is a very theoretical assumption and the employed scaling rates are not true in practice:
 - Matrix multiplication only scales O(n^3) on paper. Modern Algorithms and frameworks are highly optimized for this kind of operation, accelerating it. Moreover, modern highly-parallel Hardware (i.e. GPUs) yields approximately linear scalability with respect to the number of compute units (The A100s they use in their experiments alone provide 8192 nodes each)
- Likewise their assumption for the communication part with O(n^2) is not correct, as modern communication schemes perform operations like Allreduce (which is  the most commonly used communication for such distributed models) in O(n log(n)).
- Even if the assumptions on the Square-Cube-Principle were correct, the authors provide not thorough/rigorous experimental validation/proof of this law, but only a single measurement to demonstrate that for a few sample cases, computation is higher then communication (Fig. 1 and 3)
- One of the main components of their algorithm, the stochastic rewiring of the pipeline connections is neither explained nor demonstrated. How does this work? How do you determine the used weights? Why do you use 300s as time interval for the round robin algorithm? I would assume the length of the computation window is highly problem-dependent. How do you handle adaptive rebalancing, avoiding cascades? One can easily think of situation where all peers in one swarm are idle and hence all leave to an other stage, leaving the swarm empty? How do you avoid such deadlocks?
Also in the experiments, the proclaimed superiority of SWARM when confronted with peers leaving or joining is not tested or demonstrated. And while you show convergence in the same number of iterations as the original algorithm, not a single word is spent on actual compute time? How long does training actually run?
- One of the main motivations of their work is that highly specialized hardware is expensive and hence not available to many researchers in developing countries. Yet all experiments are run on T4 cloud devices and A100s. This is not cheap hardware! The argument is that one could use many cheap devices instead of fewer high-end ones. How many of these devices would I actually need? And what about the connections between those devices? Would the cost of the network not outweigh the saving for compute devices?


**Summary Of The Paper:**

The paper investigates the possibilities to train large model-parallel neural networks (e.g. large transformer networks) via pipelining on heterogeneous distributed hardware architectures. The rational behind their work is that full HPC environments and large-scale super-computers are not available to many researchers, especially in developing countries, and hence a method for training on cost-efficient distributed „preemtible“ hardware instances would be favorable in these cases. Specifically they employ model pipelining (distributing layers across nodes) together with random dynamic  pipeline routing and data compression, and evaluate their algorithm on training  GPT-2 on cloud T4 and A100 GPUs.

**Summary Of The Review:**

The topic studied is definitely of interest, yet the authors do not manage to sufficiently explain their method. Furthermore, they fail to demonstrate the advantages of their method that they claim. The experimental part is lacking substantial thoroughness, and more studies should be conducted in that regard.

---

> ### Author Response · Authors · 2021-11-11
> **Author Response to Reviewer RSKK (Part 2)**
>
> > One can easily think of situation where all peers in one swarm are idle and hence all leave to an other stage, leaving the swarm empty?
>
> While we have never encountered this in our evaluation, this can be resolved with a simple tie-breaking mechanism. For instance, a peer should switch if it comes up first when ordered by the tuple (stage utilization, peer throughput, peer ID), where peer ID is that peer’s unique identifier.
>
> Furthermore, in the model-parallel training setting, it is actually nontrivial to devise a situation where a given stage of the pipeline is under exactly zero load: the only cases we could think of are (1) a significantly imbalanced split of model layers across the stages (for example, 1 layer in one stage and 11 in another) or (2) a prohibitively high network latency combined with the large granularity of pipelined microbatches. The first issue is resolved by correct model partitioning (which needs to be done for any model parallel approach), the second one can be alleviated by increasing the timeouts and the utilization measurement period ($T$ in our paper).
>
> > How long does training actually run?
> For the results presented in Figure 4, the experiments took approximately 2 and 3 weeks of wall time for the baseline and the SWARM models respectively.
>
> > One of the main motivations of their work is that highly specialized hardware is expensive and hence not available to many researchers in developing countries. Yet all experiments are run on T4 cloud devices and A100s. This is not cheap hardware!
>
> As we emphasize in the abstract, Sections 1, 2, and 4.3, we use cheap preemptible instances both for cost efficiency and to be able to run reproducible experiments with unstable node uptime at scale. For instance, **the main experiment in Section 4.3 was conducted using only instances with T4 GPUs that cost $\approx$ 0.11 USD per hour**. The preemptible A100 instances used for heterogeneity evaluation cost $\approx$ 1.7 USD per GPU per hour, but they are still more cost-efficient than non-preemptible nodes from the same cloud provider, which cost over 3.4 USD per GPU per hour.
>
> > The argument is that one could use many cheap devices instead of fewer high-end ones. How many of these devices would I actually need?
> In fact, as stated in the abstract and the introduction, the argument of our work is not on using cheaper compute accelerators instead of faster ones, but on the actual possibility of model parallelism outside of uniform HPC setups. As a result, the key problems that we study in this work are the unreliability of devices in heterogeneous conditions combined with relatively slow network connectivity.
>
> > And what about the connections between those devices? Would the cost of the network not outweigh the saving for compute devices?
>
> The networking costs are small precisely because SWARM is communication-efficient. In other words, compression techniques such as quantized activations and maxout also help reduce the total communication cost along with our observations on the model size.
>
> In our specific case, we allocated all T4 instances in the same region, so the cost of communication was zero. However, if these were allocated in two cloud regions, the average networking cost would still be \\$22.16 per hour, compared to \\$68.96 per hour in compute costs.
>
> ### References
>
> [1] Understanding the efficiency of GPU algorithms for matrix-matrix multiplication. Kayvon Fatahalian, Jeremy Sugerman, and Pat Hanrahan. 2004
>
> [2] Strassen’s Algorithm Reloaded on GPUs. Jianyu Huang, Chenhan D. Yu, and Robert A. van de Geijn. 2020
>
> [3] Matrix Multiplication on High-Density Multi-GPU Architectures: Theoretical and Experimental Investigations. Peng Zhang and Yuxiang Gao. 2015
>
> [4] Bandwidth optimal all-reduce algorithms for clusters of workstations. Pitch Patarasuk, Xin Yuan. 2009
>
> [5] Efficient Large-Scale Language Model Training on GPU Clusters Using Megatron-LM. Deepak Narayanan, Mohammad Shoeybi, Jared Casper, Patrick LeGresley, Mostofa Patwary, Vijay Anand Korthikanti, Dmitri Vainbrand, Prethvi Kashinkunti, Julie Bernauer, Bryan Catanzaro, Amar Phanishayee, Matei Zaharia. 2021
>
> [6] Using DeepSpeed and Megatron to Train Megatron-Turing NLG 530B, the World’s Largest and Most Powerful Generative Language Model. Kharya, Paresh and Alvi, Ali. 2021
>
> [7] A simple and fast parallel round-robin arbiter for high-speed switch control and scheduling. Zheng, S.Q. & Yang, Mei & Blanton, John & Golla, Prasad & Verchere, Dominique. 2002

---

> ### Author Response · Authors · 2021-11-11
> **Author Response to Reviewer RSKK (Part 1)**
>
> We thank the reviewer for their feedback! We particularly appreciate the request to further explain and evaluate the adaptive components of SWARM and to include the extra reference. However, we believe that the main shortcomings of the paper outlined in the review can be addressed with a few clarifications, which we provide in our response and will incorporate in an upcoming update of the paper.
>
> > Matrix multiplication only scales O(n^3) on paper. Modern highly-parallel Hardware (i.e. GPUs) yields approximately linear scalability.
>
> The practical time complexity for multiplying $n \times n$ matrices is far from linear [1, 2, 3]. While modern GPUs may multiply *small matrices* with linear complexity, multiplying matrices with $n > 1000$ saturates the GPU cores [1,2] and even multi-GPU setups [3]. Our target setup deals with $n{=}4096$ and $n{=}12288$ while some use cases use $n{=}20480$ or even $n{=}25600$ [5,6]. We address this in Section 3.1 (second paragraph), where we describe both theoretical and practical aspects of matrix multiplication on GPUs.
>
> In addition, we also note that our analysis considers the asymptotic computational complexity, which is a fundamental constraint of the algorithm: regardless of the implementation and the hardware, parallel versions have to perform the same total amount of computation.
>
> > Likewise their assumption for the communication part with O(n^2) is not correct, as modern communication schemes perform operations like Allreduce (which is the most commonly used communication for such distributed models) in O(n log(n)).
>
> We believe that this section of the work did not clearly convey our original intent and thus caused a slight misunderstanding, so please allow us to address that. First, All-Reduce is used for aggregating gradients of parameters *in data-parallel training* and does not apply for passing activations between pipeline stages *for model-parallel training*. Second, the $O(n \times \log n)$ All-Reduce complexity ($O(n)$ for bandwidth-optimal algorithms such as Ring All-Reduce [4]) is with respect to **the number of peers**, whereas in our case $n$ is the **tensor dimension**. In other words, if a pipeline stage expects a $batch \times n_{units}$ matrix, it requires $O(batch \times n_{units})$ communication. We explain this and provide additional examples at the beginning of page 5.
>
> > Even if the assumptions on the Square-Cube-Principle were correct, the authors provide not thorough/rigorous experimental validation/proof of this law
>
> While we believe that our current experiments (Section 4.1) cover an important and practically relevant use case of training Transformers across a broad range of model configurations and network setups (see Table 1), we agree that the paper would benefit from additional evaluation. To that end, we will report an additional evaluation with computer vision models by the end of the next week.
>
> > (on stochastic rewiring) How do you determine the used weights?
>
> The weights of each worker are equal to this worker’s throughput, measured in samples per second. SWARM trainers estimate the compute throughput of peers in each pipeline stage using an exponential moving average. Each time a trainer submits a batch to a peer and receives a result, the trainer will update its estimated throughput based on the response time. On top of this, we use an interleaved weighted round-robin scheduler that uses throughput estimates to assign tasks to peers of the same stage in a balanced manner. More specifically, we use the heap-based implementation of IWRR as described in [7]. In the nearest revision, we will add a more detailed description of stochastic wiring to the appendix.
>
> > (on adaptive rebalancing) Why do you use 300s as time interval for the round robin algorithm?
>
> First, allow us to clarify that this time interval is used for the swarm rebalancing component and not the round-robin scheduler. In our preliminary experiments, we found that SWARM is not very sensitive to this parameter, at least in the 60-600s range. A good rule of thumb is to set it to approximately the same time as it takes to initialize a peer (initialize the GPU runtime and load the latest parameters).
>
> The sole purpose of this parameter is to resolve situations where the pipeline stages are imbalanced *and no new peers join the experiment*. When training with preemptible cloud instances, having a sufficiently large fleet (>50 instances) usually implies that devices are joining and leaving every few minutes. In that case, assigning new peers to the most underrepresented pipeline stage will be enough to balance it. To illustrate this argument, we will provide an additional evaluation of SWARM performance under different rebalancing parameters.

---

### Official Review · Reviewer_XtNW · 2021-11-02

**Correctness:** 4
**Technical Novelty And Significance:** 4
**Empirical Novelty And Significance:** 2
**Recommendation:** 6
**Confidence:** 4

**Main Review:**

The authors present an elegant approach for training neural networks at scale without a dedicated HPC system. It addresses an important need in the community and is likely to have a significant impact. SWARM parallelism is essentially a generalization of pipeline parallelism, with similar benefits (efficiently increasing model capacity) and downsides (communication overhead, need for micro batching, bubble overhead). The dynamic swarm rebalancing does alleviate some of the load balance issues in pipeline parallelism.

The communication overhead is especially painful in environments with high-latency, low-bandwidth communication. Reducing the proportion of time spent in communication by increasing the model size will improve the FLOP throughput, but then the authors must justify that this doesn’t hurt the truly important metrics: learning quality and time-to-solution. The baseline model in Section 4.2 is not convincing: it is about 2x larger than GPT-2 Small but achieves worse learning quality. This is further confused by the different compute budgets when training the two models. That said, it is a nice result that compressing the pipeline boundaries with maxout only resulted in modest degradations to learning quality. It will be nice to see if the authors could get comparable results to an existing Transformer model only using the maxout trick.

Finally, the large-scale training in a cloud environment is a good proof-of-concept for the effectiveness of SWARM parallelism. One can estimate the authors achieve 61% of peak throughput when training with 400 T4s and 56 A100s. However, as discussed above, the authors should demonstrate that the final model achieves a competitive learning quality in a reasonable time frame. Also, more thorough scaling studies, with and without device heterogeneity, would be helpful in understanding this algorithm’s behavior at scale.

Minor issues (mostly typo and grammar errors):
1. .....as each machine can be leave training abruptly....
2. Repeated "that that " on page 6
3. Repeated "memory memory " on page 16

**Summary Of The Paper:**

The paper describes a novel parallelism strategy for training large neural networks: compute devices are grouped into “swarms”, pipeline parallelism is applied within each swarm, and pipeline parallelism is applied between swarms. Communication between swarms is randomized and swarms are periodically rebalanced, making this approach amenable to systems with heterogeneous and/or unreliable devices of different sizes and capacities. The authors discuss two techniques to reduce the communication overhead: increasing the compute volume (since it tends to grow faster than the data volume) and modifying the model architecture (compression-aware architecture) to reduce the data volume at pipeline boundaries. The authors empirically investigate the effect of these techniques when training Transformer models and they apply SWARM parallelism to train in a cloud environment with heterogeneous GPUs.

**Summary Of The Review:**

SWARM parallelism is an elegant approach to train large neural networks in non-HPC settings and it has been shown to achieve decent compute throughput in a cloud environment. However, it requires several model modifications to run efficiently, and it has not been shown that these changes do not degrade the learning quality or slow down training.

---

> ### Author Response · Authors · 2021-11-13
> **Author Response to Reviewer XtNW**
>
> We thank the reviewer for their feedback and a comprehensive summary of our contributions. It appears that the most important request is to demonstrate SWARM performance in terms of time to solution, rather than training throughput. We outline our plan to address the reviewer's request below and plan to upload an updated version of the paper by the end of the next week.
>
> > authors must justify that this doesn’t hurt the truly important metrics: learning quality and time-to-solution.
>
> We agree wholeheartedly. In the paper, we demonstrate that SWARM produces exactly the same updates as existing distributed training algorithms and will therefore achieve a solution in the same number of steps. Since SWARM performs these steps only marginally slower (but at a significantly smaller cost) than the baseline, we argue that SWARM is more cost-efficient than strategies for training large models.
>
> Empirically, we verify this observation in Figure 4, where we demonstrate that SWARM has the same stepwise performance as ZeRO-Offload on A100 GPUs with the same hyperparameters. We traced this equivalence for the first 20B training tokens, which took 2 weeks for the A100 baseline and 3 weeks for SWARM on preemptible T4 nodes. For reference, training the baseline for Figure 4 already took us over \\$73K, compared to \\$14K for SWARM. Unfortunately, this is as far as we can go because of the baseline’s compute cost.
>
> That said, we agree with the reviewer’s request and plan to (1) run an additional experiment where we train a smaller Transformer model to convergence and (2) include a formal argument for the equivalence between SWARM and the ZeRO-Offload baseline in the appendix.
>
> > the baseline model in Section 4.2 is about 2x larger than GPT-2 Small but achieves worse learning quality
>
> The GPT-2 model was trained on a large dataset [1] which is not publicly available. In turn, we ran our experiments on OpenWebText, which is similar but not equivalent to the GPT-2 dataset. Aside from that, we did our best to replicate the original setup but failed to do so. Since training GPT-2 is very expensive and our computational resources were limited, we opted to instead train a larger model with less compute. While large models are more compute-efficient, the compute we used was not sufficient to reach the GPT-2 baseline perplexities.
>
> Despite training on a different dataset and with less compute, we opted to include GPT-2 in Table 2 to give readers a better understanding on how our experimental setup compares to existing ones. The setup was not meant to replicate GPT-2 performance. To address the reviewer's concern, we plan to clearly communicate our reasoning in Section 4.2. However, we are willing to discuss alternative solutions (e.g. removing GPT-2) if the reviewer finds it necessary.
>
> While we have strong confidence in our codebase, we will provide additional results on WikiText-103 to establish that the compression methods degrade perplexities only slightly.
>
> > demonstrate that the final model achieves a competitive learning quality
>
> The training curves for maxout compression are very similar to the baseline model and we expect these trends to continue when training further. Currently, we do not have the computational budget to perform these experiments for all models (or within the rebuttal time period). However, we are able to apply maxout to WikiText-103, which is public and easily reproducible, and we will report the performance once these results finish training.
>
> > Also, more thorough scaling studies, with and without device heterogeneity, would be helpful in understanding this algorithm’s behavior at scale.
>
> While the paper already has 3 different scaling experiments (two in Section 4.1 and another in Section 4.3), we agree that additional evaluations would be helpful. To that end, we plan to add another set of experiments for training throughput with different swarm sizes and for alternative model architectures. We hope this will alleviate the reviewer’s concern, but we are eager to discuss further evaluations if necessary.
>
> > Minor issues (mostly typo and grammar errors)
>
> Thank you! We appreciate the effort and will fix all typos by the nearest update.
>
> ### References
> [1] Language Models are Unsupervised Multitask Learners. Alec Radford, Jeff Wu, Rewon Child, David Luan, Dario Amodei, and Ilya Sutskever. 2019
>
> [2] Language Models are Few-Shot Learners. Tom B. Brown, Benjamin Mann, Nick Ryder, Melanie Subbiah, Jared Kaplan, Prafulla Dhariwal, Arvind Neelakantan, Pranav Shyam, Girish Sastry, Amanda Askell, Sandhini Agarwal, Ariel Herbert-Voss, Gretchen Krueger, Tom Henighan, Rewon Child, Aditya Ramesh, Daniel M. Ziegler, Jeffrey Wu, Clemens Winter, Christopher Hesse, Mark Chen, Eric Sigler, Mateusz Litwin, Scott Gray, Benjamin Chess, Jack Clark, Christopher Berner, Sam McCandlish, Alec Radford, Ilya Sutskever, Dario Amodei. 2020

---

> > ### Author Response · Authors · 2021-11-21
> > **Follow-up to the remark on training large models**
> >
> > > Reducing the proportion of time spent in communication by increasing the model size will improve the FLOP throughput, but then the authors must justify that this doesn’t hurt the truly important metrics: learning quality and time-to-solution.
> >
> > While we agree with the concern and plan to address it promptly, we also note that there is numerous prior work showing the advantages of training large neural networks. For instance, a recent line of work [1, 2] argues that larger autoregressive models become more efficient as we increase the compute budget. That is, increasing the parameter count under the same total number of FLOPs improves the final quality. While we already mention these results in Section 1, we can refer to them in other parts of our work if necessary, for example, in the subsection about the square-cube law or in the section covering our experiments.
> >
> > ### References
> > [1] Scaling Laws for Neural Language Models. Jared Kaplan, Sam McCandlish, Tom Henighan, Tom B. Brown, Benjamin Chess, Rewon Child, Scott Gray, Alec Radford, Jeffrey Wu, Dario Amodei. 2020
> >
> > [2] Scaling Laws for Autoregressive Generative Modeling. Tom Henighan, Jared Kaplan, Mor Katz, Mark Chen, Christopher Hesse, Jacob Jackson, Heewoo Jun, Tom B. Brown, Prafulla Dhariwal, Scott Gray, Chris Hallacy, Benjamin Mann, Alec Radford, Aditya Ramesh, Nick Ryder, Daniel M. Ziegler, John Schulman, Dario Amodei, Sam McCandlish. 2020

---

### Author Response · Authors · 2021-11-16
**General Response to Reviewers**

Dear reviewers, we thank you for taking the time to evaluate our work and for providing comprehensive feedback on it.

In particular, we would like to highlight that all reviewers appreciated the novelty and significance of the paper (**Reviewer XtNW**: “It addresses an important need in the community and is likely to have a significant impact”, **Reviewer RSKK**: “the research question at hand is definitely valuable and of importance”, **Reviewer 8gAI**: “The paper addresses an important problem that can ... make the training accessible to more researchers”, **Reviewer 5Pxy**: “The proposed SWARM framework is novel and practical”), agreeing that our work offers a simple yet effective approach for solving an important problem. Reviewers 8gAi and 5Pxy also listed the clarity of writing and the broad coverage of related work as the strong points of our paper.

While there were questions regarding the details of a few non-key aspects of the work, we addressed them in our responses to reviewers and took this feedback (along with extra references) into account when preparing a new version of the paper, which is available now in OpenReview.

The notable changes that we made include:
* **A more detailed description of stochastic wiring** in Appendix D, as requested by Reviewers RSKK and 5Pxy;
* **An explanation of the relation between SWARM and ZeRO-Offload** and **equivalence of SWARM and existing distributed methods** in Appendix E, as requested by Reviewers XtNW, 8gAi and 5Pxy;
* **Additional surveys on model-parallel training** in the background section, as requested by Reviewer RSKK;
* **Small notation and typo fixes**, as noted by Reviewers XtNW, 8gAi, and 5Pxy.

In addition, the reviewers asked us to provide more empirical results on the comparison with regular distributed training, as well as the experiments showing the necessity of both stochastic wiring and adaptive rebalancing. We are currently running these experiments and will do our best to provide the initial results this week; in the meantime, we encourage you to read our responses and tell us if your other concerns have been resolved.

---

> ### Comment · Reviewer_8gAi · 2021-11-18
> **Follow-up #2**
>
> I agree that this research direction is important. Furthermore, the updated version clarifies missing details.
> > However, even in the updated version, there is not a single speedup experiment.
>
> To assess the impact of the different suggested components of the SWARM algorithm I require a speedup breakdown. Furthermore, a fair comparison (all experiments in the same settings) to previous work is required. Due to the time limitations, simply measuring the algorithms' throughput (not full training) is sufficient as convergence experiments with final accuracy have already been provided.

---

### Author Response · Authors · 2021-11-21
**General Response to Reviewers (Update 2)**

Dear reviewers, we have uploaded an updated version of our paper that contains several experiments that were requested during the discussion phase. In particular,
* Appendix F contains **detailed performance comparisons of SWARM parallelism with GPipe and Offload** both in high-latency and low-latency settings, as requested by Reviewer 8gAi;
* Appendix G contains **the evaluation of adaptive rebalancing** based on historical data, as requested by Reviewers RSKK and 5Pxy;

The experiments on the necessity of stochastic wiring, its scaling with the number of nodes and models for other DL problems are currently in the works. If you have any questions or remarks regarding the setup of currently reported or upcoming experiments, feel free to ask us for details or adjustments.

---

> ### Comment · Reviewer_8gAi · 2021-11-21
> **Follow-up #3**
>
> Thank you for the updated results and clarifications.
> I'll update my review accordingly.

---

### Author Response · Authors · 2021-11-26
**General Response to Reviewers (Update 3)**

Dear reviewers, an updated version of the paper is available on OpenReview. It has the following additions:
* Appendix H gives more experiments on **comparing the convergence** of SWARM with the baseline methods (both for distributed training algorithms and for compressions schemes) in terms of wall time and total iterations, as suggested by Reviewers XtNW and 8gAi;
* In Appendix I, we evaluate **the scaling properties of SWARM** with respect to the number of peers at a single stage to provide more insight into its performance, as requested by Reviewer XtNW.

We also investigated how the compute and communication requirements scale for vision models to provide additional evidence for the square-cube law. For this investigation, we consider CoAtNet — the current state-of-the-art architecture for ImageNet classification. More specifically, we took three models: CoAtNet-2, CoAtNet-7 from the original paper and a model based on CoAtNet-7, but with all layer dimensions scaled by a factor of 4x. The three models have S4 dimensions of 1024, 3072 and 12288, respectively. For each model, we assign CoAtNet’s S0, S1 and S2 to the pipeline stage 1, while COAtNet’s S3 and S4 are computed by stage 2. We measure the device utilization for each model on 512x512 images, based on the original configuration for CoAtNet-7. At 1Gb/s interconnect, the average GPU utilization was 18.1\% for CoAtNet-2, 26.6\% for CoAtNet-7 and  61.3\% for the largest model, which replicates the trend for GPT-3 layers in Figures 1 and 3.

If there are any unresolved questions, we would be happy to answer them in a discussion.

[1] CoAtNet: Marrying Convolution and Attention for All Data Sizes. Zihang Dai, Hanxiao Liu, Quoc V. Le, Mingxing Tan. 2021

---

### Decision · Program_Chairs · 2022-01-20

**Decision:**

Reject

**Comment:**

Overall, the reviewers thought this paper suggested an important problem.  However, there were many concens.  Particularly, the multiple reviewers felt it was unclear when the new approach is better than prior work. The reviewers had difficulty connecting the experiments to the paper's main claims.